# Hormone Replacement Therapy and Cardiovascular Health in Postmenopausal Women

**DOI:** 10.3390/ijms26115078

**Published:** 2025-05-24

**Authors:** Wenhan Xia, Raouf A. Khalil

**Affiliations:** Vascular Surgery Research Laboratories, Division of Vascular and Endovascular Surgery, Brigham and Women’s Hospital, Harvard Medical School, Boston, MA 02115, USA; xwh19881230@ncu.edu.cn

**Keywords:** endothelium, estrogen, progesterone, testosterone, vascular smooth muscle

## Abstract

Sex-related differences are found not only in the reproductive system but also across various biological systems, such as the cardiovascular system. Compared with premenopausal women, cardiovascular disease (CVD) tends to occur more frequently in adult men and postmenopausal women (Post-MW). Also, during the reproductive years, sex hormones synthesized and released into the blood stream affect vascular function in a sex-dependent fashion. Estrogen (E2) interacts with estrogen receptors (ERs) in endothelial cells, vascular smooth muscle, and the extracellular matrix, causing both genomic and non-genomic effects, including vasodilation, decreased blood pressure, and cardiovascular protection. These observations have suggested beneficial effects of female sex hormones on cardiovascular function. In addition, the clear advantages of E2 supplementation in alleviating vasomotor symptoms during menopause have led to clinical investigations of the effects of menopausal hormone therapy (MHT) in CVD. However, the findings from these clinical trials have been variable and often contradictory. The lack of benefits of MHT in CVD has been related to the MHT preparation (type, dose, and route), vascular ERs (number, variants, distribution, and sensitivity), menopausal stage (MHT timing, initiation, and duration), hormonal environment (progesterone, testosterone (T), gonadotropins, and sex hormone binding globulin), and preexisting cardiovascular health and other disorders. The vascular effects of sex hormones have also prompted further examination of the use of anabolic drugs among athletes and the long-term effects of E2 and T supplements on cardiovascular health in cis- and transgender individuals seeking gender-affirming therapy. Further analysis of the effects of sex hormones and their receptors on vascular function should enhance our understanding of the sex differences and menopause-related changes in vascular signaling and provide better guidance for the management of CVD in a gender-specific fashion and in Post-MW.

## 1. Introduction

Cardiovascular disease (CVD), for example, hypertension (HTN), coronary artery disease (CAD), and stroke, is among the most common and costly diseases, causing ~30% of mortality in the United States and worldwide [1,2]. CVD involves changes in different vascular tissues and cells, including endothelial cells (ECs), vascular smooth muscle cells (VSMCs), and extracellular matrix (ECM). Genetics, age, and sex differences are major predisposing factors in vascular dysfunction and CVD [3]. In this context, the terms “sex” and “gender” should be carefully interpreted. “Sex” is a genetically defined state among mammals, dependent on the X and Y chromosomes and supported by the gonadal hormones. Sex assignment as male, female, or intersex usually occurs at birth based on the appearance of the external genitalia and is verified by analysis of the chromosomes and gonads. Thus, sex differences involve genetic, biological, and hormonal factors [4,5]. On the other hand, “gender” is a psycho-social state among humans, involving the person’s internal sense of belonging to certain societal categories as a man, woman, or nonbinary individual. Therefore, gender-related variations may be influenced by psychosocial factors and are subject to change over time [6,7].

Variations in sex chromosome genes influence the course of CVD [3,8]. The sex hormones estrogen (E2), progesterone (P4), and testosterone (T) also act on specific vascular receptors to stimulate steroid hormone-responsive genes and signaling pathways that reinforce the sex differences in CVD. The decline in sex hormones’ levels with age could influence vascular function and the course of CVD. Although CVD occurs less frequently in premenopausal women (Pre-MW) compared to adult men, its progression accelerates after menopause in postmenopausal women (Post-MW), ultimately surpassing the incidence observed in men [9,10]. Mechanistic studies have also shown marked effects of E2, P4, and T on ECs, VSMCs, ECM, and various proteases. Additionally, menopausal hormone therapy (MHT) improves menopausal vasomotor symptoms in Post-MW [11]. These findings have led to clinical trials investigating the effects of MHT on CVD; however, the outcomes have remained inconclusive [12], making it important to investigate the causes of MHT failure and further examine the vascular effects of sex hormones.

This review focuses on the effects of sex hormones on vascular function and the benefits/risks of MHT on cardiovascular Health and in Post-MW. As a narrative review, our goal was to synthesize key perspectives rather than exhaustively include all the relevant literature. The review cites 478 articles, selected following three steps: (1) Database search: The PubMed database was searched for relevant articles published between 2000–2024 using the keywords “hormone replacement therapy”, “menopausal hormone therapy”, and “cardiovascular health”. This initial research yielded ~820 articles. (2) Screening: Careful screening excluded non-English articles, case reports, and small-sample-size research without justification. On the other hand, when a recent article cited original and seminal research reports from previous years, these reports were included. This narrowed the search to 600 articles for further review. (3) Final inclusion: 478 articles were cited based on relevance, originality, impact, expert input, and novelty. The cited articles represent the most influential and evidence-supported research work in this field.

We first describe the synthesis and metabolism of sex hormones and the changes in their levels with age and their association with CVD. We then describe the effects of E2 and estrogen receptor (ERs) on vascular function. We describe the benefits of MHT for menopausal symptoms and the benefits versus risks of MHT in CVD. We also discuss the factors contributing to MHT failure in clinical trials, including the MHT preparation (type, dose, and route), vascular ERs (number, variants, distribution, and sensitivity), menopausal stage (MHT timing, initiation, and duration), hormonal environment (P4, T, gonadotropins, and sex hormone binding globulin), and preexisting cardiovascular health and other disorders. We conclude with some suggestions to enhance the vascular benefits of MHT on cardiovascular health in Post-MW and highlight the impact of using E2 and T supplementation in gender affirming therapy and their differential effects in cis- and transgender individuals.

## 2. Synthesis and Menopause-Related Changes of Sex Hormones

Sex hormones are primarily produced through the steroidogenic conversion of cholesterol in the gonads, but they are also synthesized in other tissues, such as the adrenal glands, adipose tissue, skeletal muscle, breast, and liver (Figure 1). Estrogens have a cyclopentanoperhydrophenanthrene nucleus, the basic ring structure of all steroids. Natural estrogens including estrone (E1), estradiol (E2), and estriol (E3) have four rings, namely A, B, C, and D; a hydroxyl group (OH) at C3; and an OH or a ketone group at C17 [13]. In Pre-MW, E2 is predominantly produced by the ovaries and shows maximal estrogenic activity. E1 can be generated in the liver through the conversion of E2 and is also derived from androstenedione in peripheral tissues. E1 is weaker than E2 because it lacks the 17β-OH group but could influence estrogenic activity when converted to E2. E3 is a weak estrogen with low affinity and activity on ERs [14].

Approximately 2–3% of endogenously secreted estrogens remain in an unbound, biologically active form. Due to the estrogens’ small molecular size and high lipophilicity, they are capable of rapid and widespread distribution. E2 has an estimated half-life of around 3 h and undergoes dynamic interconversion with E1. Natural estrogens are primarily metabolized in the liver through several enzymatic pathways, including oxidation and hydroxylation by cytochrome P450 enzymes, glucuronidation mediated by UDP-glucuronosyltransferase, sulfation via sulfotransferase, and O-methylation catalyzed by catechol O-methyltransferase [15]. E2 metabolism varies with polymorphisms in the genes encoding steroidogenic enzymes, race, age, menstrual cycle stage, menopausal status, enzyme modulators, and cigarette smoke [16].

Menopause is an age-related physiological transition defined by the absence of menstruation for 12 consecutive months. It is marked by diminished ovarian function, reduced levels of E2 and P4 secretion, and a modest yet consistent production of T [17,18] (Figure 1), causing relative increase in the T/E2 ratio (Figure 2). Also, while ovarian production and circulating levels of E2 and E1 are decreased, increased aromatase and 17β-HSD2,4 activity in adipose tissue leads to increased peripheral conversion of androgens to E1 and relative increase in circulating E1/E2 ratio in Post-MW [19]. In Post-MW, the main source of dehydroepiandrosterone, a precursor of estrogens and androgens, is the adrenal glands, with the ovaries contributing ~20% [20,21]. Also, in Post-MW, ~75% of androgens are synthesized in peripheral tissues from dehydroepiandrosterone, and ~25% of circulating T is secreted by the ovaries [22]. The non-aromatizable and more potent T derivative dihydrotestosterone (DHT) is markedly decreased or almost undetectable in Post-MW [23]. Early loss of ovarian function, whether due to primary ovarian insufficiency or elective bilateral oophorectomy, performed to lower the risk of ovarian cancer, can have a significant impact on women’s health. Although endocrine activity in postmenopausal ovaries is minimal, bilateral oophorectomy after menopause has been linked to reduced concentrations of E2, dehydroepiandrosterone [24,25], and serum T [17,20]. Among women aged 50 to 89 years, total bioavailable T levels were found to be 40–50% lower in those who had undergone oophorectomy compared to those with intact ovaries [26]. These findings indicate that postmenopausal ovaries continue to serve as an important source of androgens in older women, and the long-term implications of reduced T levels following oophorectomy warrant further investigation.

Steroid hormones are subject to metabolic processes and interconversion of their precursors and metabolites, which can affect both the absolute and relative levels of these hormones. This, in turn, may influence sex differences in CVD and vascular function and modulate the vascular effects of sex hormones.

## 3. Menopausal Symptoms and Menopause-Related CVD

Menopausal changes in sex hormone levels are linked to a range of vascular, metabolic, physical, and emotional symptoms that negatively impact the quality of life in Post-MW. Approximately 50% of women experience vasomotor and psychological symptoms, such as hot flushes, night sweats, sleep disturbances, stress, and mood swings, which begin during the menopausal transition and may persist for around 4.5 years [29,30]. The relationship between vasomotor symptoms and CVD varies depending on their type, severity, and the woman’s age. Women with hot flushes or night sweats are often found to have additional CVD risk factors, such as reduced E2 levels, heightened sympathetic and catecholaminergic activity, and elevated calcitonin gene-related peptide levels [31,32,33,34]. Sleep disturbances and mood fluctuations also contribute to adverse cardiovascular outcomes, increasing the risk of arterial stiffness and vascular dysfunction [35]. In the SWAN study of midlife women, depression was found to be associated with the progression of coronary artery calcification, a risk comparable to that linked with high body mass index (BMI) and systolic blood pressure (BP) [36]. Severe vasomotor symptoms, whether occurring before or after menopause, are linked to an elevated risk of CVD [37,38]. Among women <60 years, the risk of a cardiovascular event is higher in women with vasomotor symptoms than without. However, there is no difference among women >60 years [39]. In addition, hot flushes and night sweats show distinct associations with cardiovascular markers in women during early postmenopause [40]. These findings indicate that variable hormonal profiles contribute to the manifestation of menopausal symptoms and related cardiovascular risk, pointing to a highly individualized menopausal trajectory that uniquely influences the development of menopause-related CVD and its responsiveness to MHT [41].

Evidence from studies on sex differences supports a link between reduced ovarian function and an increased risk of CVD. The divergence in CVD risk between sexes begins as early as childhood and progresses along distinct life-course trajectories, shaped by genetic dimorphism and hormonal influences [42]. Adult males are more prone to developing ischemic heart disease and CAD compared to females, which often results in heart failure with reduced ejection fraction (HFrEF) and elevated mortality rate at a younger age [43,44,45]. Conversely, older women more frequently present with inflammation, increased vascular stiffness, and concentric left ventricular hypertrophy, predisposing them to heart failure with preserved ejection fraction (HFpEF) [46], with different disease progression and response to medications [47]. The onset of menopause, typically occurring around age 52, aligns with a notable rise in CVD prevalence [48]. Post-MW show a higher incidence of CAD and atherosclerosis compared to Pre-MW [49,50]. Although CAD develops several years later in women versus men [48], the gender gap in the CAD mortality rate ratio in men versus women narrows from 4–5 in middle age (30–64 years) to 2 at 65–89 years [51]. Since 1984, women have had a higher CVD mortality rate than men, and by the year 2000, around 60,000 more women than men succumbed to CVD. Also, while age-adjusted mortality from CAD has declined by roughly 50% since peaking in the 1960s, this reduction has been significantly less pronounced in certain populations, particularly among women, non-Caucasians, and those of lower socioeconomic status [52,53,54]. Postmenopausal sex hormone profiles are also closely tied to the elevated CVD risk observed in older women. Specifically, a higher T/E2 ratio in Post-MW correlates with increased incidence of CVD, CAD, and heart failure, with elevated T levels linked to greater risk and higher E2 levels associated with reduced CAD risk [55]. Additionally, BP tends to be lower in Pre-MW compared to adult men, but among Post-MW aged 65 years and above, the prevalence of HTN surpasses that in men [56,57] (Figure 2). Among the elderly, the number of women with HTN is twice that of men [58]. A study in China also showed that lower socioeconomic status was associated with a greater incidence of HTN and that women were more susceptible [59]. Women with HTN are also more likely than men to have vascular and myocardial stiffness at old age and more often have isolated systolic HTN, reflecting increased aortic stiffness [60]. Importantly, at the same level of cuff systolic BP, invasive aortic systolic BP in women is 4.4 mmHg higher than in men. This suggests that cuff BP may not be as effective in women as in men for identifying CVD risk associated with systolic BP and pulse pressure, raising the possibilities of underdiagnosis and missed treatment opportunities to prevent CVD events [61]. This may also explain why women have a higher CVD risk than men at the same cuff BP [62,63]. Taken together, these observations indicate that HTN significantly contributes to the heightened cardiovascular event risk observed in women and that conventional cuff-based BP measurements may underestimate this CVD risk [61]. Furthermore, the timing of menopause onset affects cardiovascular health. Women who experience early or premature ovarian insufficiency face elevated risks of developing HTN, aortic stenosis, CAD, heart failure, atrial fibrillation, ischemic stroke, and venous thromboembolism (VTE) and are more likely to experience increased overall mortality [49,64,65]. For both natural and surgical menopause, there is a graded relationship between menopausal age and CVD incidence. Studies have found that for every 1 year decrease in menopausal age, the incidence of CAD increases by 2% [66,67]. Compared with women with an average age of natural menopause, women with premature and early natural menopause have a markedly increased risk of experiencing a first non-fatal CVD, CAD, or stroke before the age of 60 [50]. Also, of the nine biomarkers were found to be positively associated with early menopause, five with an increased risk of future heart failure (adrenomedullin, adipsin, β2M, N-terminal pro-B-type natriuretic peptide (NT-proBNP), and C-reactive protein (CRP)), and four with major CVD (adipsin, β2M, NT-proBNP, and CRP), suggesting that distinct biological pathways may make women with early menopause more susceptible to CVD [68]. Bilateral oophorectomy leads to a much more abrupt loss of ovarian steroids than natural menopause, causing higher rates of HTN, CAD, heart failure, and VTE [69]. The risk of CVD was greatest in women who underwent surgical menopause at a younger age, especially <40 years, compared with women who had natural menopause at 50–54 years [67]. Therefore, elective bilateral oophorectomy should be discouraged at the time of hysterectomy surgery for benign disease, as it may increase CVD risk [70].

Distinct sex-based trajectories in CVD progression point toward a potential role of reduced ovarian E2 in its underlying pathophysiology. Biomarkers proposed for identifying premature ovarian insufficiency include shortened telomere length and reduced telomerase activity in granulosa cells [71]. Evidence from studies on Post-MW and individuals with polycystic ovary syndrome (PCOS) further supports the link between lower ovarian E2 levels and CVD [47,72]. Also, aromatase is an enzyme known to convert androgens into estrogens, and when it is pharmacologically inhibited in E2-dependent cancers, such inhibition has been associated with elevated cardiovascular risk [73]. Studies of specimens in the UK biobank revealed that higher levels of free circulating E2 correlated with reduced risk of heart failure in men [74]. However, the cardioprotective effects of E2 appear to wane in women after menopause, with a marked increase in heart failure incidence observed beyond 55 years of age [75,76]. Notably, pulmonary arterial hypertension (PAH) occurs across both sexes and all ages. In contrast to the male dominance in general CVD prevalence, familial PAH occurs more frequently in women, who also exhibit superior right ventricular function, improved prognosis, and higher survival —collectively known as the “E2 paradox”— possibly due to alterations in ERs and associated vascular pathways [77,78]. Also, stroke prevalence in women exhibits a unique trend [79]. Between the ages of 25–45 years, women have a higher incidence than men, and this trend persists in postmenopause, although the gender gap diminishes with advancing age, with men eventually equaling or surpassing women at very old age [80,81] (Figure 2).

## 4. Sex Differences in Vascular Function

The regulation of vascular tone is modulated by the functional interplay between ECs and VSMCs as well as the equilibrium between intrinsic and extrinsic mechanisms and diverse vasoconstrictive and vasodilatory stimuli. Among extrinsic regulators, neurohumoral influences are predominant, such as sympathetic nerve input and neurotransmitters, circulating catecholamines (epinephrine and norepinephrine), angiotensin II (Ang II), vasopressin, and 5-hydroxytryptamine (serotonin), most of which enhance vascular tone. Conversely, certain circulating agents like atrial natriuretic peptide exert vasodilatory effects, thus reducing tone. Intrinsic regulation includes myogenic responses originating from VSMCs, which contribute to increased tone, as compared to endothelium-derived relaxing factors like nitric oxide (NO), prostacyclin (PGI_2_), and endothelium-derived hyperpolarizing factor (EDHF). Contracting factors from ECs include endothelin-1 (ET-1) and thromboxane A2 (TXA_2_). Additionally, locally acting hormones and chemicals such as histamine and bradykinin may either constrict or dilate vessels, while hypoxia and metabolic by-products typically reduce vascular tone. Extrinsic mechanisms primarily affect systemic vascular resistance and BP, whereas intrinsic pathways enable microvasculature to adapt to fluctuations in intraluminal pressure and help fine-tune regional tissue perfusion and blood flow [82]. The active component of myogenic tone depends on intracellular Ca^2+^ signaling and the function of mechanosensitive Ca^2+^ channels in VSMCs [83], whereas passive components rely on structural proteins and enzymes involved in vascular remodeling and are Ca^2+^-independent.

Evidence from both human and animal studies demonstrates sex-based differences in vascular tone across various vascular regions and in response to extrinsic vasoconstrictors. For instance, the vasoconstrictive response to norepinephrine is significantly attenuated in the forearms of women compared to men. Calcitonin gene-related peptide, a factor likely involved in migraine pathophysiology, induced smaller relaxation and was less potent in middle meningeal arteries isolated from women versus men <50 years but showed no sex difference among ≥50 years groups [84]. In coronary arteries of pigs, oxidized low-density lipoprotein (LDL) was shown to amplify 5-hydroxytryptamine-induced contractions more significantly in males compared to females [85]. Also, in vivo studies revealed that male rats exhibited a more pronounced pressor response to vasopressin infusion than their female counterparts [86]. In rat aortas, phenylephrine triggers a stronger contractile response in males compared to females. This response does not differ between castrated and intact males, but it is significantly heightened in ovariectomized (OVX) females relative to intact females, indicating that variations in vascular tone between sexes are likely influenced by E2 levels and the activity of vascular ERs [87]. Sex-related distinctions have also been observed in intrinsic mechanisms governing vascular tone. For instance, differences have been documented in the regulation of myogenic tone and the contractile response to ET-1 in coronary resistance vessels of rats [88]. Moreover, in small murine arteries, modulation of myogenic tone by NO differs by sex, with EDHF contributing to enhanced vasodilation only in female vessels [89].

Variations in vascular function associated with sex and menopause are linked to distinct alterations in different components of the blood vessels. Endothelium-dependent brachial artery flow-mediated dilation (FMD) is highest in Pre-MW (~9.9%) and decreases progressively in perimenopause (early 8.2%, late 6.5%) and Post-MW (early 5.5%, late 4.7%) [90]. Measurements of forearm blood flow showed reduced vasodilator response to brachial artery infusion of acetylcholine (ACh) in hypertensive men and Pre-MW and Post-MW versus normotensive subjects, with a negative correlation with advancing age among all subjects. Among normotensive women, endothelium-dependent vasodilation was normal in Pre-MW but impaired in Post-MW. Among hypertensive subjects, decreased EC function with age was less in Pre-MW than in men but similar in Post-MW and men, suggesting that the protective effects of E2 on EC function decline in early menopause and worsen with prolonged E2 deficiency in Post-MW [90,91]. In Chinese women, arterial stiffness has been shown to increase more rapidly from mid-life into the postmenopausal phase, indicating that menopause may drive alterations in structural proteins and ECM [92]. Also, studies in chemically induced menopausal mice have demonstrated that cerebral arterioles exhibit heightened myogenic tone, attributed to reduced hyperpolarization of VSMCs, which is linked to diminished activity of large-conductance Ca^2+^-activated K^+^ channels (BK_Ca_) [93].

## 5. Estrogen Receptors (ERs) in the Vasculature

E2 interacts with different vascular ER subtypes, including ERα, ERβ, G-protein coupled ER (GPER), and several ER variants [94,95]. The *ESR1* gene on chromosome 6q25.1 encodes ERα, while *ESR2* on chromosome 14q23-24.1 encodes ERβ [13]. Human ERα and ERβ have an overall 44% homology and share the domain structure observed in the nuclear receptor superfamily, i.e., A/B, C, D, E, and F domains (Figure 3). The A/B domain harboring the activation function 1 (AF-1) is involved in protein–protein interactions and transcriptional activation of target genes. The central C domain representing the DNA-binding domain (DBD) contains four cysteines arranged in two zinc fingers and is involved in DNA binding and receptor dimerization. DBD is highly conserved in ERα and ERβ and shares 95% amino acid identity such that both ERs bind the same E2 response elements (EREs), recognize similar DNA sequences, and regulate similar target genes. The D domain, or hinge domain, is not well conserved between ERα and ERβ (30%), and it facilitates ER association with heat-shock protein 90 (HSP90) and its nuclear localization. The E domain or ligand-binding domain (LBD) has ~55% homology between ERα and ERβ and harbors a ligand-dependent AF-2 that facilitates ligand binding and receptor dimerization. The F domain contains co-factor recruitment regions [15].

The LBDs of ERα and ERβ have similar affinities for E2. The LBD has twelve helices (H1–H12) folded into a 3-layered, anti-parallel, α-helical sandwich comprising a central core layer of α-helices H5, H6, H9, and H10 sandwiched between two additional layers of α-helices H1–4 and H7, H8, and H11, with H12 at the bottom [15]. This helical arrangement creates a wedge-shaped molecular scaffold and a sizeable ligand-binding cavity at the narrower end of LBD, into which the hormone binds. The binding of E2 with ER typically involves non-covalent H-bonding and additional hydrophobic contacts and interactions.

ERα and ERβ have been identified in the female reproductive system, the heart, blood vessels, lungs, brain, and bones, causing various effects on the cardiovascular, immune, nervous, and musculoskeletal systems [96,97]. ERα knockout (KO), ERβ KO, and ERα/ERβ double-KO mice survived but showed reproductive malfunction [98]. E2 binds to ERs with high affinity and specificity. E2 diffuses through the plasma membrane and complexes with cytosolic/nuclear ERα and ERβ, which bind to chromatin and promote transcription of genes with EREs, thus inducing genomic effects and altering vascular cell growth [99,100] (Figure 4). E2 also interacts with plasmalemmal ERα and ERβ to form homodimers or ERα/ERβ heterodimers, activate protein kinases and signaling pathways, and induce rapid non-genomic vascular effects [100,101,102,103,104]. ERα and ERβ have been detected in ECs [105,106,107] and are also expressed in VSMCs of the human coronary artery, aorta, iliac artery, and saphenous vein, with greater number in females versus males [108]. ERα was found in the rat uterine artery, and ERβ was abundant in the rat aorta, tail, and uterine arteries [109]. ERα was found to mediate many of the E2 vascular protective effects [110]. Also, E2-mediated cardioprotection against ischemia/reperfusion injury is mimicked by ERα agonists and unaffected by ERβ antagonists [111]. In macrophages, E2 via ERα inhibited inflammatory gene expression and lipopolysaccharide (LPS)-induced DNA binding, transcriptional activation, and nuclear translocation of p65 nuclear factor-κB (NFκB) [112]. In ascending aortic arteriosclerosis specimens from Post-MW, the endothelium showed a decrease in ERα expression and autophagy but increased inflammation and pyroptosis. Also, Post-MW showed increased macrophage activation and plaque instability with increased MCP-1, matrix metalloproteinase-9 (MMP-9), TLR4, MYD88, and NF-κB p65 and decreased ERα and TIMP-1 expression in the vascular endothelium. OVX female LDLR^−/−^ mice also showed decreased serum E2 levels and arterial ERα expression and collagen content, in association with arterial macrophage accumulation, increases in plaque and necrotic core areas, and upregulated TLR4 and MMP-9 expression. E2 or phytoestrogen upregulated ERα and increased plaque stability by inhibiting macrophage accumulation and TLR4 signaling. In RAW264.7 cells, LPS decreased ERα and TIMP-1 expression and increased TLR4 activation, and these effects were reversed by E2 or phytoestrogen, and the E2 reversal was abolished by the ERα antagonist methylpiperidinopyrazole (MPP). Thus, decreased ERα expression promotes macrophage infiltration and plaque instability postmenopause, and activation of ERα alleviates atherosclerotic plaque instability by inhibiting TLR4 signaling and macrophage-related inflammation [113]. Also, in the aortas of OVX female ApoE^−/−^ mice and homocysteine-treated HUVECs, E2 accelerated autophagy and ameliorated inflammation and cell pyroptosis, and these effects were reversed by inhibition/silencing of ERα [114]. Other studies showed greater expression of ERβ than ERα in human coronary arteries, particularly in VSMCs [115]. In rats, ERβ mRNA levels increased in response to aortic balloon injury [116]. Also, E2 activation of ERβ modulated tumor necrosis factor-α (TNFα)-induced inflammatory response in rat aortic VSMCs [117]. The process of E2/ER-mediated gene transactivation involves the translocation of E2/ER complex to the nucleus, where it binds to DNA and the ERE on specific genes, resulting in enhanced gene transcription. A study comparing the effects of ERα and ERβ on gene transactivation involved co-transfecting ER-negative HeLa cells with ERE2-TATAtk-CAT or vitellogenin (VIT-tk-CAT) promoter/reporter constructs, along with increasing amounts (1 to 100 ng) of ERα or ERβ expression vectors. After 24 h of treatment with 10 nM E2, chloramphenicol acetyltransferase (CAT) transcription driven by each promoter was measured. In E2-stimulated ER-transfected HeLa cells, ERα acted as a stronger gene transactivator at low ER concentrations, but as the ER concentration increased, ERα activity self-suppressed, and ERβ emerged as the stronger transactivator [108].

In addition to the wild-type ERα, there are five ERα variants that feature specific deletions in the exons encoding regions of the hormone-binding domain. These variants lack exon 4 (ERαΔ4), exon 5 (ERαΔ5), exon 6 (ERαΔ6), exon 7 (ERαΔ7), and both exons 6 and 7 (ERαΔ6,7) [118]. The expression of ERα splicing variants differs across various tissues, influencing certain facets of ERα signaling, and in some cases, these variants may even counteract the effects of full-length ERα [119]. Transient expression studies in HeLa cells demonstrated that increasing levels of ERαΔ7 progressively inhibited E2-induced transcriptional activation by both wild-type ERα and ERβ on ERE-driven promoters [120]. Variants of ERβ have also been identified, and further research is needed to understand their biological roles [118]. Investigating the function of ER variants in normal physiology as well as the changes in their expression, activity, and interactions with each other and wild-type ERs could provide valuable insights into the mechanisms that govern the shifts in E2 effects across different stages of a woman’s life. Estrogens interact with various ERs, each with distinct potencies for transactivating ERE-containing gene promoters and mediating cellular responses [121]. In comparison with E2 (K_d_ 0.1 nM), E1 (K_d_ 0.3 nM) and E3 (K_d_ 1.4 nM) have a lower affinity for ERα and display different cardiovascular effects [122,123]. Human ECs show varying responses to E2 and E1, with the E2/E1 ratio playing a critical role in determining the overall estrogenic effects [124].

GPER, also known as GPR30, is a 7-transmembrane G protein-coupled ER that plays a role in mediating rapid non-genomic effects [125]. GPER has been detected in various cellular regions and tissues, including the plasma membrane and endoplasmic reticulum of the brain, peripheral tissues, as well as the human mammary artery and saphenous vein [126,127,128]. In studies on mice, female mice showed reduced BP, pulse wave velocity, and arterial stiffness, with a more pronounced carotid artery ACh-induced relaxation compared to males. However, these sex differences were absent in GPER-deficient mice, highlighting the differential impact of nongenomic E2/GPER signaling on the vascular phenotype of males and females [129]. Also, treatment with the GPER agonist G1 was found to enhance vascular function and lessen myocardial ischemia/reperfusion injury in aging female rats [130]. Furthermore, GPER1 is involved in mediating protective cardiovascular and renal effects that help regulate BP, positioning GPER1 as a promising target for HTN management [131].

## 6. Cardiovascular Effects of E2

E2 is the primary sex hormone in women, playing a key role in the development of the reproductive system, maintaining metabolic balance, regulating energy production and distribution, promoting mitochondrial biogenesis, and facilitating adenosine triphosphate (ATP) synthesis [132]. In addition, E2 via nuclear ERs and GPER activates multiple cardiovascular effects on different vascular cells and biological processes (Table 1). For instance, E2 decreases the susceptibility to vascular inflammation, atherosclerosis, and cardiovascular events [133]. E2 promotes anti-inflammatory effects by decreasing neutrophil infiltration of the vascular wall, oxidative stress, and the expression of monocyte chemoattractant protein-1 and TNFα [111,117,134]. The decline of E2 levels during menopause impairs mitochondrial function and increases inflammation, pathologic angiogenesis, and microvascular disease [135]. E2 offers several vascular advantages, including alterations in circulating lipoproteins, prevention of lipoprotein oxidation, regulation of homocysteine levels, modulation of blood coagulation and fibrinolytic factors, and suppression of collagen accumulation in blood vessels [87,136]. Additionally, E2 may influence BP by inhibiting sympathetic vasoconstriction, regulating the renin–angiotensin–aldosterone system (RAAS), modulating salt sensitivity, controlling body mass, and reducing oxidative stress, all of which contribute to vascular inflammation as well as cardiovascular and renal damage and HTN [137]. In an OVX female 1-kidney deoxycorticosterone acetate (DOCA)-salt treated rat model of HTN, E2 replacement prevented DOCA-salt induced HTN and the underlying increases in GABAergic excitation in hypothalamic magnocellular arginine-vasopressin (AVP) neurons and subsequent increases in AVP release and plasma levels, likely through modulation of the expression/activity of NKCC1 (Cl^−^ importer) and KCC2 (Cl^−^ extruder) [138]. Importantly, E2 directly modulates vascular tone, causing various degrees of vasodilation in conduit vessels, small arteries, and microvessels through differential activation of ERs in ECs, VSMCs, and ECM [139].

### 6.1. Effects of E2 on ECs

ECs detect fluid shear stress and circulating cytokines, thereby participating in the regulation of blood flow, vasodilation and vasoconstriction, vascular permeability, macromolecule extravasation, blood fluidity, as well as platelet and leukocyte activity [209]. Elevated shear stress and enhanced sympathetic activity, oxidative stress, and vascular inflammation contribute to EC dysfunction, HTN, atherosclerosis, and platelet aggregation. OVX female ApoE^−/−^ mice fed a high-fat diet showed accelerated atherosclerosis and increases in ferroptosis markers, plasma lipid peroxidation and iron deposition in the atherosclerotic plaque. Treatment with E2 or ferroptosis inhibitor ferrostatin-1 alleviated atherosclerosis, lipid peroxidation, and iron deposition and upregulated the cystine/glutamate antiporter (xCT) and glutathione peroxidase 4 (GPX4) in ECs. In ECs with oxidized-LDL or Erastin-induced ferroptosis, E2 promoted anti-ferroptosis effects through antioxidative mechanisms, improved mitochondrial function, and upregulated GPX4 expression, and these effects were attenuated by inhibition of nuclear factor erythroid 2-related factor 2 (NRF2), suggesting a role of EC ferroptosis in postmenopausal atherosclerosis progression and E2 protection against EC ferroptosis via the NRF2/GPX4 pathway [210]. ECs secrete both vasodilators like NO, PGI_2_, and EDHF and vasoconstrictors such as ET-1 and TXA_2_, along with vascular endothelial growth factor (VEGF). The binding of E2 to cytosolic or nuclear ERs initiates genomic responses, resulting in the phosphorylation of mitogen-activated protein kinase (MAPK, ERK1/2) and stimulation of EC proliferation [211]. In human ECs, E2 enhanced angiogenesis, wound healing, tube formation, ERK1/2 phosphorylation and adenosine, and decreased extracellular ATP and adenosine deaminase activity, and these effects were reversed by blocking ER or the adenosine receptor, suggesting that E2 modulates angiogenesis via ER and adenosine-mediated pathways [135]. In healthy Post-MW, infusion of E2 improved FMD and enhanced ACh-induced vasodilation in the forearm [212]. Also, infusion of E2 in dogs as well as in isolated rat and rabbit hearts increased coronary blood flow [213]. Additionally, ACh caused greater relaxation in the aorta of female versus male spontaneously hypertensive rats (SHRs) [214]. E2 facilitates vasodilation by enhancing the release of relaxing factors such as NO, PGI_2_, and EDHF while reducing the release of vasoconstrictors like ET-1 and TXA_2_ [215].

NO is a major vasodilator produced from the transformation of L-arginine to L-citrulline by neuronal NO synthase (nNOS and NOS I), inducible NOS (iNOS and NOS II), and endothelial NOS (eNOS and NOS III) [216]. While eNOS is Ca^2+^-dependent and rapidly regulates vascular tone, iNOS is Ca^2+^-independent and regulates vascular function in the long term. NO production is greater in Pre-MW versus men [217]. In microvessels isolated from subcutaneous fat biopsies, FMD was found to be lower in Post-MW compared to Pre-MW but showed improvement when Post-MW vessels were exposed to E2 or the ERα agonist propyl pyrazole triol (PPT) [218]. Additionally, E2 enhanced both basal and stimulated NO release in human coronary artery ECs. In OVX female guinea pigs, E2 supplementation elevated EC-derived NO production and increased coronary artery responsiveness to vasodilators [219]. Moreover, E2 contributes to vasodilation in pulmonary arteries by promoting EC NO synthesis [220]. Furthermore, arteries from female rats exhibit higher EC NO release than those from males, primarily attributed to elevated E2 levels in females [214,221].

E2-induced release of NO from ECs is partially mediated through the activation of nuclear ERs, engagement of genomic signaling pathways, and the subsequent upregulation of eNOS [105,221]. E2 promotes angiogenesis, upregulates VEGF and eNOS in human coronary ECs, and prevents destabilization of eNOS mRNA by TNFα [222,223]. E2 also enhances eNOS activity and vasodilation by activating EC plasmalemmal ERs and non-genomic pathways [221,224] (Figure 4). In resting conditions, eNOS is tightly bound to caveolin within the caveolae of the EC plasma membrane. Upon E2/ER activation, there is an elevation in cytosolic free Ca^2+^ concentration ([Ca^2+^]_c_), initiated by Ca^2+^ release from the endoplasmic reticulum and maintained through Ca^2+^ influx via store-operated channels [225]. Additionally, E2 acting through GPER may suppress plasma membrane Ca^2+^-ATPase (PMCA) function in ECs, thereby extending the duration of the Ca^2+^ signal. In support of this mechanism, PMCA and GPER have been shown to physically interact and modulate each other’s activity. Overexpression of GPER1 or its selective activation by G1 has been demonstrated to reduce PMCA function, limiting Ca^2+^ extrusion and increasing [Ca^2+^]_c_ [226]. The rise in [Ca^2+^]_c_ subsequently binds calmodulin, and this Ca^2+^/calmodulin complex facilitates the dissociation and translocation of eNOS from plasma membrane caveolin. E2 also activates MAPK and the PI_3_K/Akt pathway, promoting the phosphorylation and relocalization of cytosolic eNOS to the plasma membrane, where it undergoes myristoylation/palmitoylation, leading to its full activation [227,228]. In support, membrane-impermeant E2 binds plasmalemmal ERs and stimulates NO release from human ECs [227]. Also, E2 treatment in OVX female mice induces rapid NO production and vasodilation [229]. NO produced in response to E2 stimulation diffuses into adjacent VSMCs, where it activates guanylate cyclase, resulting in the generation of cyclic guanosine monophosphate (cGMP). Functioning as a second messenger, cGMP subsequently activates protein kinase G (PKG), which lowers VSMC [Ca^2+^]_c_ by enhancing the activity of both PMCA and the sarcoplasmic/endoplasmic reticulum Ca^2+^-ATPase (SERCA). In addition, PKG phosphorylates and suppresses myosin light-chain kinase (MLCK), thereby reducing the sensitivity of actin–myosin filaments to [Ca^2+^]_c_ and promoting VSM relaxation [230].

The role of ERα and ERβ in mediating NO release has also been examined [106]. Transfer of ERα gene into bovine aortic ECs was shown to upregulate eNOS expression [231]. In COS-7 cells co-transfected with ERα and eNOS, rapid activation of eNOS by E2 was successfully reconstituted [232]. Basal NO levels were reduced in ERα KO mice, indicating a predominant function of ERα in this process [233]. ERα-selective agonists were found to improve EC dysfunction in blood vessels of OVX female SHRs [234]. In human ECs, a truncated ERα isoform lacking the N-terminal region (ERα46) interacts with c-Src, PI_3_K, Akt, and eNOS within EC caveolae, contributing to E2-mediated activation of eNOS [235]. Other studies in COS-7 cells have shown that overexpression of ERβ enhanced its association with plasma membrane caveolae and E2-induced eNOS activity independently of ERα [236]. Moreover, the vasodilatory response to E2, along with the activation of MAPK and PI_3_K signaling pathways, is not observed in mice lacking both ERα and ERβ (double knockout) [229], suggesting that both ERα and ERβ are involved in NO production. The GPER agonist G1 also increases endothelial NO production through phosphorylation/activation of eNOS. In human EA.hy926 ECs and HUVECs, G1 increased eNOS and Kruppel-like factor-2 (KLF2) expression, and these effects were suppressed by silencing GPER or KLF2 or inhibition of G_αq_ and G_βγ_. G1 also induced Ca^2+^ signaling and increased CaMKKβ, CaMKIIα, and AMPK phosphorylation/activity, which in turn promoted phosphorylation of histone deacetylase 5 and transcriptional KLF2, and the expression of eNOS, supporting a role of GPER in regulating eNOS levels [237].

E2 also has antioxidant effects that enhance NO bioactivity. Superoxide anion (O_2_^•−^) levels are greater in the aorta of male versus female rats [238]. In OVX female rats, protracted E2 deficiency causes activation of RAAS, increased BP, oxidative stress, plasma lipoperoxides and free radicals, and reduced levels of antioxidants and NO bioavailability [239], while E2 replacement decreases free radicals and increases nitrites/nitrates levels [240]. E2/ER complex translocates to the mitochondria, where it regulates mitochondria-encoded genes [136]. E2 also inhibits NADPH oxidase expression and the generation of reactive oxygen species (ROS), O_2_^•−^, H_2_O_2_, and peroxynitrite (ONOO^−^); reduces oxidative stress; and enhances NO bioactivity [241,242], and some of these effects may involve downregulation of Ang II type 1 receptor [243]. Also, in HUVECs, E2 reverses H_2_O_2_-induced oxidative stress, cell senescence, and apoptosis by suppressing thrombospondin-1 expression and consequent TGF-β/Smad signaling [244].

PGI_2_ is an endothelium-derived relaxing factor synthesized through arachidonic acid metabolism via cyclooxygenase enzymes COX-1 and COX-2. Upon binding to prostanoid receptors on VSMCs, PGI_2_ activates adenylate cyclase, leading to the production of cyclic adenosine monophosphate (cAMP). The rise in cAMP subsequently activates protein kinase A (PKA), which reduces VSMC [Ca^2+^]_c_ and diminishes myofilament sensitivity to Ca^2+^ through mechanisms similar to those employed by cGMP/PKG, ultimately resulting in VSM relaxation. E2 enhances ACh-induced dilation in subcutaneous vessels of Post-MW partly through COX-2 activation [245]. Also, in arteries from OVX female monkeys with experimentally induced atherosclerosis, PGI_2_ levels were found to be diminished, whereas local E2 treatment of the vessels enhanced PGI_2_ synthesis [246]. E2 increases urinary excretion of PGI_2_ metabolites in ERβ but not ERα KO mice, supporting a role for ERα in PGI_2_ production [247]. In other studies, E2 upregulated COX-1 and promoted PGI_2_ synthesis in fetal ovine pulmonary artery and mouse cerebral artery ECs through ERβ [221,248,249]. In contrast, the relaxation effect of E2 in rabbit coronary arteries remained unchanged following COX inhibition by indomethacin, indicating that prostanoids play only a limited role in this response [250]. E2 may trigger a vasodilator crosstalk such that an increase in NO would decrease COX-mediated effects [251].

The greater vascular relaxation in female versus male vessels may also involve EDHF and K^+^ channels [252]. ACh-induced hyperpolarization and relaxation of mesenteric arteries are smaller in male and OVX female versus intact female rats, and the differences are eliminated by K^+^ channel blockers. E2 replacement in OVX female rats improves ACh-induced vascular hyperpolarization and relaxation [253,254]. The effects of E2 on ECs vary along the arterial tree, with NO more dominant in large conduit arteries, while resistance vessels produce more EDHF [255].

E2 cardiovascular effects may also involve non-genomic enhancement of endothelial hydrogen sulfide (H_2_S) release. In HUVECs, E2 induced phosphorylation of cystathionine γ-lyase, the key enzyme in H_2_S generation. E2 enhanced the interaction of membrane ERα with Gαi-2/3 to transactivate particulate guanylate cyclase-A, produce cGMP, and activate PKG-Iβ, leading to phosphorylation of cystathionine γ-lyase and rapid release of H_2_S. Silencing of cystathionine γ-lyase or particulate guanylate cyclase-A in mice attenuated E2-induced aortic dilation, supporting a role of H_2_S as an E2/ERα-activated pathway [256]. E2/ER-induced release of H_2_S may also activate ATP-sensitive K^+^ channels (K_ATP_) in VSMCs, thus promoting vascular hyperpolarization and relaxation [257].

E2 through ER also increases hemeoxygenase(HO)-2 activity and carbon monoxide (CO) production in HUVECs and human uterine artery ECs. Interestingly, low E2 concentrations (10^−10^ M) induced CO-mediated activation of soluble guanylyl cyclase and a 56% increase in cellular cGMP levels. By contrast, higher E2 concentrations had no effects, likely due to NOS-mediated NO release and inhibition of CO release, supporting crosstalk between E2/ER-induced vasodilator factors [258]. In ECs, CO also activates BK_Ca_ channels, resulting in ECs hyperpolarization and subsequent vascular wall hyperpolarization, diminished reactivity to vasoconstrictors, and vascular relaxation [259,260,261].

The sex differences in vascular tone also involve endothelium-derived contracting factors such as ET-1 and TXA_2_. ET-1 stimulates ET_A_R and ET_B2_R in VSMCs to induce contraction and ET_B1_R in ECs to cause vascular relaxation. In Post-MW with CAD, intracoronary administration of E2 decreases ET-1 in coronary sinus plasma [262]. ET-1 release, vascular tone, and BP are reduced in female versus male SHR [252,263]. In DOCA-salt hypertensive rats, mesenteric arteries of males exhibit stronger contraction to ET-1 than those of females, likely due to variations in ET_B_R expression. The expression of ET-1 and ET_B2_R as well as the vasoconstriction induced by the ET_B_R agonist IRL-1620, is more pronounced in mesenteric arteries of OVX females compared to intact females, with E2 replacement in OVX females reversing this effect, suggesting that E2 reduces ET-1 and ET_B2_R expression [264]. Additionally, aortic strips from male rats subjected to trauma hemorrhage show enhanced ET-1-induced contraction, which is reduced by E2 or the ERβ agonist diarylpropionitrile (DPN) treatment [265]. E2 also inhibits both basal production and ET-1 release from ECs stimulated by serum, TNFα, transforming growth factor-β1, Ang II, and thrombin [266,267]. EC release of the COX-derived vasoconstrictor TXA_2_ is also higher in male compared to female SHR [252].

### 6.2. Effects of E2 on VSMCs

Proliferation and migration of VSMCs contribute to the progression of intimal hyperplasia and atherosclerosis. While E2 promotes EC growth, it suppresses VSMC proliferation by inhibiting MAPK and activating protein phosphatases [268], stimulating the release of endothelial NO and subsequent increase in cGMP in VSMCs or increasing cyclic adenosine monophosphate (cAMP) and adenosine [269]. Also, E2-induced H_2_S protects against atherosclerosis by upregulating Kelch-like ECH-associated protein 1 (Keap1) expression and inhibiting VSMC calcification and by downregulating MMP-2 and α5β1 integrin expression and inhibiting VSMC proliferation/migration and vascular inflammation [270]. In cultured rat aortic VSMCs, E2 binding to ERα reduced cell proliferation by 50% and increased the expression of zinc finger transcription factor KLF4 and shifted it to the nucleus, partly through activation of PI_3_K, Akt phosphorylation, and induction of NOS, and these effects required KLF4 and manganese superoxide dismutase (MnSOD) as shown by transfection experiments. In a carotid artery injury model, MnSOD^+/−^ mice showed more extensive neointima formation and Ki67 immunostaining than wild-type (WT) mice. E2 deficiency led to enhanced neointima formation and increased Ki67-positive proliferating cells in OVX WT and MnSOD^+/−^ mice. E2 replacement prevented neointima formation in WT but not MnSOD^+/−^ mice. Cultured VSMCs derived from MnSOD^+/−^ mice showed enhanced proliferation as compared to WT VSMCs, and E2 treatment failed to inhibit proliferation in MnSOD^+/−^ VSMCs, supporting a role of E2 in inhibiting VSMC proliferation through MnSOD and KLF4, thus providing potential targets for treatment of vascular restenosis [271]. Sirtuin 1 (SIRT1) is a class III histone deacetylase involved in vascular metabolism, atherosclerosis, and aging. Aortas from OVX mice showed marked VSMC hyperplasia and upregulation of SIRT1, which was reversed by E2 supplementation. Also, treatment of A7r5 VSMCs with E2 downregulated SIRT1 expression, increased apoptosis, and reduced proliferation, viability, and migration through Akt and ERK pathways, and these effects were reversed by resveratrol-induced activation of SIRT1, suggesting that E2 inhibits VSMC proliferation through regulation of SIRT1 expression by Akt and ERK pathways [272]. In human and porcine coronary artery SMCs, G1-induced activation of GPER inhibited cell proliferation, promoted differentiation, and increased α-actin and smooth muscle protein 22α (SM22α) [273]. Also, 2-methoxy-E2 inhibited VSMC proliferation and pulmonary vascular inflammation [274], showing similar efficacy to sildenafil and bosentan in alleviating monocrotaline-induced PAH in both male and female rats. Additionally, when combined with sildenafil or bosentan, 2-methoxy-E2 provided greater protection, reversed established PAH, and improved survival rates [275]. The distinct effects of E2 on EC versus VSMC growth likely depend on the specific ER subtype or the ERE present in the promoters of target genes. For example, E2 primarily activates ERα to upregulate miR-203, which suppresses VSMC proliferation while also enhancing the expression of miR-143 and miR-145, which promote communication between ECs and VSMCs [276].

E2 through activation of plasmalemmal ERs also rapidly inhibits VSMC contraction in endothelium-denuded rabbit and porcine coronary arteries [161,277]. The relaxing effects of E2 depend on the vascular bed, estrous cycle, and prior exposure to E2. For example, the largest relaxation in response to E2 is observed in the tail and mesenteric arteries of female rats during the pro-estrous phase. Additionally, E2 induces less microvascular relaxation in OVX+E2 compared to OVX rats, indicating that chronic exposure to E2 may reduce the expression of ERs [278]. Notably, the rapid ex vivo E2-induced vascular relaxation requires micromolar concentrations several times higher than the in vivo picomolar serum levels. As estrogens are lipophilic, their tissue concentrations are higher than those in the serum. Furthermore, higher concentrations of E2 may be necessary to trigger relaxation in the aorta, which contains multiple VSMC layers and dense collagen, unlike smaller microvessels with thinner media and fewer VSMCs.

VSMC contraction involves an increase in [Ca^2+^]_c_ and myosin light-chain phosphorylation. Additionally, protein kinase C (PKC) and rho-kinase (ROCK) inhibit myosin light-chain phosphatase, enhancing the sensitivity of the contractile myofilaments to [Ca^2+^]_c_. VSMCs from the rat aorta are longer and have lower basal [Ca^2+^]_c_ in females compared to males and in OVX+E2 females compared to OVX females [279]. In VSMCs incubated in a Ca^2+^-free solution, phenylephrine or caffeine induced small contractions and [Ca^2+^]_c_ levels that were similar in both male and female rats whether intact or gonadectomized, suggesting no difference in intracellular Ca^2+^ release. However, in VSMCs exposed to a Ca^2+^-containing solution, phenylephrine or high KCl caused greater increases in contraction and [Ca^2+^]_c_ in intact males and OVX females compared to intact or OVX+E2 females, pointing to sex differences in Ca^2+^ influx [279]. Supporting this finding, the expression of L-type Ca^2+^ channels is elevated in coronary VSMCs of male swine compared to females and in cardiomyocytes from ER-deficient mice [280,281]. E2 also inhibits PGF_2α_- and KCl-induced sustained VSM contraction, Ca^2+^ influx, and [Ca^2+^]_c_, supporting its role in blocking Ca^2+^ channels [161,162]. Additionally, E2 activates BK_Ca_ channels in coronary VSMCs, leading to hyperpolarization and inhibition of voltage-gated Ca^2+^ channels [171]. Furthermore, E2 blocks Ca^2+^ channels in A7r5 and aortic VSMCs [282]. E2 could also reduce VSMC [Ca^2+^]_c_ by stimulating the plasmalemmal Ca^2+^ pump PMCA [168]. In rats, E2 attenuates endoplasmic reticulum stress and myocardial ischemia/reperfusion injury by upregulating SERCA2a [195]. Also, in male rat cardiac myocytes, E2 and G1 decreased Ca^2+^ transient, L-type Ca^2+^ current and sarcomere shortening and increased NO production, and these effects were reversed by the GPER antagonist G36, NOS inhibitor L-NAME, or PI_3_K inhibitor wortmannin, suggesting GPER-mediated negative inotropic effect through the PI_3_K-NOS-NO pathway [283].

Sex differences in vascular contraction may also be influenced by PKC and ROCK. Phorbol esters activate PKC, leading to a stronger contraction in the aorta of male rats compared to females. Also, the activation and translocation of α- and δ-PKC from the cytosol to the plasma membrane induced by phorbol esters are more pronounced in the aorta of male and OVX female rats compared to intact or OVX+E2 female rats, suggesting that E2 modulates PKC activity [173]. Furthermore, E2 treatment reduces the mRNA expression of ROCK in human coronary VSMCs [175].

### 6.3. E2 and ECM

The adventitia contains fibroblasts and ECM proteins such as collagen and elastin that are regulated by various proteases such as matrix metalloproteinases (MMPs). Sexually dimorphic and E2-regulated miRNAs affect EC mesenchymal transition and vascular fibrosis. E2 maintains ECM flexibility by regulating collagen and elastin turnover, modulating collagen synthesis/degradation, and maintaining the balance necessary for vascular elasticity and preventing fibrosis [284]. In OVX female rats subjected to stress induced by isoproterenol subcutaneous injections, E2 replacement reduced weight gain and BP and ET-1 expression, upregulated eNOS expression, facilitated vasomotor function, and prevented excessive intima-media thickening and collagen deposition in the aortic wall [285]. Also, prolonged treatment of female rat aorta with E2 caused increases in MMP-2 and MMP-9 gelatinolytic activity, suggesting E2-induced regulation of vascular remodeling [184]. E2 also promotes VSMC migration and the formation of varicose veins of the lower extremities, likely due to upregulation of MMP-2 and MMP-9 expression through the classical ER-mediated genomic pathways [286]. In human airway smooth muscle cells, TNFα or PDGF increased ECM deposition and MMP activity, which were reversed by the ERβ agonist WAY-200070 but not the ERα agonist PPT. WAY-200070 also decreased the activation of NF-κB and AP-1, suggesting a role of ERβ in regulating ECM production and deposition through suppression of NF-κB activity [287].

E2/ERs also regulate cardiac fibroblasts and distinctly affect cardiac fibrosis in males versus females [288]. E2 inhibits differentiation, and collagen synthesis in TGF-β1 stimulated cardiac fibroblasts by regulating cell division cycle 42 (Cdc42) and the downstream p21 (RAC1)-activated kinase 1 (Pak1)/JNK/c-Jun signaling pathway [289]. In OVX female mice, surgical induction of aortic constriction caused severe myocardial hypertrophy, fibrosis, and increased expression of the autophagy marker LC3II, and these effects were attenuated by the GPER agonist G1. Also, in H9c2 cells, G1 attenuated Ang II-induced hypertrophy and reduced LC3II expression, autophagy level, and Akt/mammalian target of rapamycin (mTOR) pathway [290]. G1 also reduced cardiac hypertrophy, fibrosis, atrial natriuretic peptide, brain natriuretic peptide and β-myosin heavy chain, and the augmented oxidative stress and apoptosis induced by Ang II in mice, whereas GPER deficiency exacerbated the Ang II effects [291], suggesting potential benefits of G1 in alleviating cardiac hypertrophy and fibrosis in Post-MW.

## 7. E2-Based MHT and CVD

E2 is administered both as a contraceptive and as part of MHT to alleviate symptoms experienced during menopause, including vaginal dryness, vasomotor symptoms, hot flushes, night sweats, and disturbances in sleep [11]. MHT is also an effective treatment for genitourinary syndrome of menopause. Given that E2 facilitates vasodilation and alleviates menopausal symptoms, MHT was initially believed to reduce the incidence and severity of HTN and CVD in Post-MW. Early observational findings from the Nurses’ Health Study (NHS) indicated a decreased risk of cardiovascular events and CVD-related mortality in Post-MW who received oral E2 [292,293] (Table 2). Among women undergoing elective coronary angioplasty, those using MHT experienced fewer cardiovascular events (12% versus 35%) and exhibited higher survival rates (93% versus 75%) compared to nonusers [294]. Nonetheless, findings from randomized clinical trials (RCTs) such as the Women’s Health Initiative (WHI) and the Heart and Estrogen/progestin Replacement Study (HERS) did not confirm cardiovascular protection and even reported an elevated risk of cerebrovascular events associated with MHT [295,296]. Both HERS and its follow-up study HERS II, along with pooled analyses, revealed no significant reduction in either primary CAD or secondary cardiovascular outcomes in Post-MW randomized to MHT versus placebo. Although a temporary decline in CAD events was observed in the MHT group during the later years of HERS, this benefit did not extend through long-term follow-up, ultimately leading to the conclusion that MHT should not be prescribed to reduce cardiovascular risk in Post-MW with existing CAD [297]. Additional concerns emerged regarding a potential association between MHT and increased VTE risk [298,299]. Furthermore, MHT use was linked to approximately a 35% higher risk of cerebral ischemia and stroke, as reported in the NHS [300], WHI [301], and the Women’s Estrogen for Stroke Trial (WEST) [302]. These observations have generated extensive research interest in the postmenopausal changes in estrogen regimens and formulations, ERs, cardiovascular health, and other biological systems as potential causes of lack of MHT benefits in CVD. The failure of MHT to elicit vascular benefits in RCTs could be related to the MHT type, dose, metabolism, route of administration, time of initiation and duration, the participants’ age, age-related changes in ER and post-ER signaling pathways, the pre-existing CVD and other clinical disorders, or the hormone environment and interaction with gonadotropins, SHBG, P4, and T (Figure 5).

### 7.1. MHT Type and CVD

Oral estrogens like conjugated equine estrogen (CEE) and E2 exhibit reduced bioavailability due to hepatic first-pass metabolism, during which they are transformed into E1 prior to entering the systemic circulation. While most MHT clinical trials have utilized oral estrogen formulations, mechanistic investigations tend to evaluate the direct vascular actions of E2. Certain synthetic estrogens, including diethylstilbestrol (K_d_ 0.04 nM) [122], esterified E2, and ethinyl-E2, demonstrate stronger binding affinity for ERα and may offer superior cardiovascular protection. Among Post-MW, use of esterified E2 has been associated with a reduced incidence of ischemic stroke and myocardial infarction (MI), without an observed increase in the risk of VTE when compared to CEE [298]. Tibolone is a synthetic steroid hormone with estrogenic, progestogenic and androgenic properties used for treatment of menopausal vasomotor symptoms and osteoporosis, but its cardiovascular benefits versus risks need to be examined [319].

Selective estrogen receptor modulators (SERMs) such as raloxifene, tamoxifen, and idoxifene are non-steroidal molecules that bind ERs and elicit purely estrogenic, partial estrogenic, or anti-estrogenic effects. SERMs decrease total cholesterol, low-density lipoprotein cholesterol (LDL-C), fibrinogen, interleukin-6 (IL-6), and TNFα. In the Raloxifene Use for The Heart (RUTH) trial, raloxifene treatment for ~5.6 years reduced the risk of invasive breast cancer in Post-MW with CVD but did not affect the risk of coronary events and increased the risk of stroke and VTE [310]. The Multiple Outcomes of Raloxifene Evaluation (MORE) study found that a 4-year course of raloxifene treatment had no impact on the incidence of cardiovascular events or stroke, although it was associated with a heightened risk of VTE [320]. PPT, which is a selective ERα agonist with 400-fold greater potency for ERα compared to ERβ, was shown to enhance FMD in the mesenteric vasculature of female mice [321]. Diarylpropionitrile (DPN), a selective ERβ agonist with 30–70 times higher potency for ERβ over ERα, induces rapid vasodilation through an NO-dependent mechanism [139]. Also, selective activation of GPER by G1 causes greater cardiovascular protection in OVX females than intact female or male rats [322]. Estrogenic agents that exhibit greater selectivity for specific ERs may improve the cardiovascular benefits associated with MHT in CVD.

Phytoestrogens, including flavonoids, isoflavonoids, stilbenes, and lignans, interact with ERs to exert estrogenic effects. Large quantities of food containing phytoestrogens, such as soybeans, red clover, wheat, and peas, are consumed by populations with lower rates of menopausal symptoms, osteoporosis, cancer, and CVD, including Japan, China, and Southeast Asia. In a double-blind RCT including Post-MW, supplementation with 56 g soy protein powder daily reduced mean, systolic, and diastolic BP compared with placebo group [323]. Other studies using soy-derived products in Post-MW showed a slight or no benefits on BP [324,325,326]. Further evaluation of the estrogenic effects of phytoestrogens is needed through well-conducted RCTs.

### 7.2. MHT Dose and CVD

The negative outcomes of MHT observed in RCTs may be attributed to the high doses used. Estrogens, being lipophilic, may have circulating levels that do not accurately represent their concentration within the vascular wall. Furthermore, E2 plays a role in adipose tissue metabolism, and a deficiency of E2 in Post-MW is linked to an increase in visceral fat. Adipose tissue exhibits P450 aromatase activity, converting androstenedione to E1, which elevates E1 levels in Post-MW with obesity. Additionally, adipose tissue releases IL-6 and TNFα, which influence vascular ERs signaling. Therefore, while a standard E2 dose in Pre-MW may be physiologically appropriate, it could lead to supraphysiological levels and adverse effects in Post-MW. CEE is available at low dose of 0.3 to 0.45 mg, standard dose of 0.625 mg, and high dose of 1.25 mg [327]. The Women’s Health, Osteoporosis, Progestin, Estrogen (HOPE) trial demonstrated a more favorable benefit/risk ratio with lower-dose CEE for preventing bone loss and alleviating vaginal atrophy, vasomotor symptoms, and endometrial bleeding [328]. The high E2 dose could also promote vascular inflammation. E2 administration in female mice for 3 consecutive months caused aortic inflammation similar to Takayasu arteritis. Also, treatment of mouse aortic VSMCs with high E2 concentrations decreased expression of contractile phenotypic markers and increased expression of macrophage-like phenotypic markers. This phenotypic switch was blocked by tamoxifen and KLF4 inhibitors and enhanced by Von Hippel–Lindau/hypoxia-inducible factor-1α (HIF-1α) interaction inhibitors, suggesting that E2-regulated phenotypic conversion of VSMCs to macrophages through Von Hippel–Lindau/HIF-1α/KLF4 axis is involved in E2-induced vascular inflammation associated with MHT use [329]. E2 also affects ECM remodeling through modulation of MMPs. Low E2 doses may inhibit MMPs and decrease collagen deposition [182], whereas high E2 doses activate MMPs and may promote vascular lesion formation and plaque destabilization [183,184]. In VSMCs from human coronary and umbilical arteries, supraphysiological concentrations of E2 elevated MMP-2 levels, which could impact vascular structure and integrity [183]. The degradation of ECM and instability of atherosclerotic plaques induced by MMPs could account for the association between alterations in the levels of MMP-2, MMP-9, and MMP-10 in Post-MW undergoing MHT and the heightened risk of cardiovascular events [330].

### 7.3. MHT Route of Administration and CVD

MHT can be administered orally or as a percutaneous gel, transdermal patch, subcutaneous implant, and vaginal preparation. Oral and transdermal estrogens formulations have similar efficacy [331]. Low-dose vaginal MHT shows little absorption into the systemic circulation and could be an option for Post-MW for whom systemic MHT is contraindicated, particularly women with a history of E2-dependent cancers, CVD, stroke, or VTE [332]. A nationwide Finnish cohort study including 195,756 Post-MW during the period between 1994 and 2009 found that vaginal MHT could reduce the risk of death from CAD and stroke in all age groups, with the highest risk reduction in women aged 50–59 years [333].

The route of administration could influence the effects of MHT on the cardiovascular system. Oral MHT undergoes first-pass metabolism and could produce benefits in the lipid profile, such as reduction in LDL-C and lipoprotein(a) and increases in high-density lipoprotein cholesterol (HDL-C), but it could increase triglycerides levels and promote thrombosis [334]. Also, despite the favorable effects of oral E2 on LDL-C, lipoprotein(a), and HDL-C levels, these subclinical benefits have not translated into reduction in CVD events or mortality among MHT users [48]. Transdermal E2 is absorbed directly into the bloodstream, bypassing the plasma E2 spikes associated with oral forms, and is converted to E1 at a rate comparable to that observed during the menstrual cycle, resulting in fewer side effects compared to oral estrogens [335]. In contrast with oral MHT, which increases triglycerides by 5% to 15%, transdermal MHT decreases triglyceride levels by 5% to 30% [336,337]. Also, because transdermal MHT preparations avoid the hepatic effect, they have less impact on coagulation factors and hemostasis [338] and a lower risk of VTE and stroke compared with oral MHT [339]. While oral E2 increases C-reactive protein in the vessel wall and may cause VTE [340], transdermal E2 increases brachial artery FMD without affecting C-reactive protein [341] and shows lower incidence of VTE [342].

Hormone replacement therapy using silastic and bioabsorbable hormone implants has gained considerable interest as a practical and effective route for treating menopausal symptoms and reduced sexual desire disorders. A retrospective study of 1,204,012 subcutaneous hormone implants during the time period between 2012 and 2019 including 376,254 men and women patients, 85% of whom were women, with 46% of them postmenopausal women, found that subcutaneous hormone implants were safer than other routes of administration of bioidentical hormone replacement therapy for both men and women [343]. Also, Pre-MW and Post-MW who reported comparable hormone deficiency symptoms showed similar improvement in total score of validated Health Related Quality of Life and Menopause Rating Scale as well as psychological, somatic and urogenital subscale scores when using T implant therapy, with better effects in women with severe complaints and greater improvement with higher T doses [344]. Nevertheless, based on relevant and recent research, the cardiovascular risks and safety of subcutaneous hormone implants for Post-MW remain to be thoroughly examined [345].

### 7.4. MHT and Menopausal Changes in ERs

The absence of vascular benefits from MHT in RCTs may also result from age-related, prolonged E2 deficiency, leading to alterations in ER expression, distribution, integrity, and post-ER signaling (Table 3). Polymorphisms in ERα at positions c.454–397 T>C (PvuII) and c.454–351 A>G (XbaI) have been associated with CAD severity in Post-MW [346]. Post-MW with the recessive ERα IVS1-397 polymorphism C/C genotype show greater HDL-C and apolipoprotein A-1 levels, brachial artery FMD, and endothelium-dependent dilation when compared to the dominant phenotype [347]. Also, among Post-MW with CAD treated with E2 or E2+P4, women with the ERα IVSI-401 polymorphism C/C genotype show two times greater increase in HDL-C levels when compared to control [306]. Age-related alterations in ER subtypes have been observed across various tissues. For example, ER is present in the retina of Pre-MW but not in men or Post-MW [348]. Also, menopause is linked to a decline in ERα expression and activity in the brain and liver, which impacts cognition and metabolism [349]. In female mice, aging leads to a reduction in ERα mRNA expression in the heart, while GPER mRNA levels rise [350]. Splice variants of ERα and ERβ, each with distinct signaling pathways, may influence vascular responses to MHT [97,118]. Full-length ERα (ERα66) and its truncated forms, such as ERα46, ERα36, and ERα30, may play distinct roles in regulating vascular function [351]. ERα46, which is localized to and dynamically targeted to the plasma membrane, enhances eNOS phosphorylation more effectively than membrane-bound ERα66 [352]. ERα36, which forms dimers with wild-type ERα66, can inhibit its activity, while ERα30, although not yet fully defined, may suppress ERα66 expression in breast cancer cells [353,354,355]. Research comparing early versus late E2 supplementation on the common carotid artery of OVX senescence-accelerated mice showed that delayed initiation of E2 supplementation increased ERα36 expression (associated with aging) without affecting ERα66 or ERα46 levels and in tandem with the negative effects of E2 [356]. Epigenetic modifications and methylation of CpG islands in the ERβ gene promoter also influence ERβ expression, vascular aging, and atherosclerosis [357]. Additionally, changes in ERα expression, E2 sensitivity, and signaling pathways modulate MHT effects differently in early (≤5 years) versus late Post-MW (>5 years past the final menstrual cycle) [358]. While ERα activation is linked to NO production and vascular protection [233], ERβ expression correlates with atherosclerotic plaque area in males [359], and the age-related shift in the ERβ/ERα ratio may promote a pro-oxidative response to E2 [360]. Interestingly, human arterioles from discarded adipose tissue showed higher EC expression of ERβ and GPER in women under 40 compared to those over 40 and in men, with ERα primarily localized in the adventitia. This finding has suggested that E2/ERβ fosters a more oxidative environment in microvascular ECs. Episodic high E2-induced oxidative stress may trigger protective mechanisms, and fluctuations in E2 levels in women not using MHT could allow time for recovery, whereas continuous MHT exposure may prevent recovery, leading to increased oxidative stress and vascular damage [255]. In ECs from peripheral veins, ERα expression was lower during the early follicular phase (lower E2) compared to the late follicular phase (higher E2) in Pre-MW and was even lower in Post-MW. ERα levels were positively correlated with eNOS expression, ser1177 phosphorylation, and endothelium-dependent brachial artery FMD, which were reduced in Post-MW, suggesting that fluctuations in estrogen status during the menopausal cycle and age-related E2 deficiency modulate EC ERα levels and vasodilatory effects [361]. Consistent with this concept, prolonged E2 deficiency in OVX female Sprague–Dawley rats is linked to a decrease in endothelial ERα expression and disrupted ERα/eNOS signaling in the aorta [362]. Aging also alters the subcellular distribution of ERα between the nucleus and cytoplasm in cholinergic neurons of mice [363]. To further test whether vascular ERs expression is reduced with aging, vascular reactivity and ERs were measured in the aorta, kidney, and heart of male and female mice of different ages. ERα and GPER were the predominant receptors in all tissues, whereas ERβ was mainly detectable in the kidney. Female mice showed higher ERα and GPER mRNA expression in the aorta and the heart. Aging affects ER transcript in a tissue-dependent manner. E2- and G1-induced vasodilation and GPER levels were reduced in the aorta of aging female mice. In the heart, ERα transcript decreased with aging, while GPER expression increased, suggesting differential impact of aging on cardiovascular ER expression in a sex- and tissue-specific pattern [350]. A separate study found no significant difference in ER expression between the aortas of aging and adult OVX SHR [364]. A deeper investigation into age-related changes in the expression of both full-length and truncated ER forms, their signaling pathways, and interactions with each other as well as with wild-type ERs, especially following menopause, could provide insights into the reasons for the absence of vascular benefits from MHT in Post-MW.

### 7.5. MHT Timing and Duration

Due to age-related changes in the vasculature, the timing of initiation and duration of MHT relative to menopause onset may influence cardiovascular outcomes [377]. Women tend to live longer, with an average life expectancy of 82.2 years, compared to men at 76.6 years [378]. Aging is linked to decline in cardiovascular function; EC senescence and dysfunction; increased VSMC proliferation, vascular remodeling, inflammation; and the development of atherosclerosis, all of which contribute to altered vascular responses to E2. As individuals age, reduced NO release and elevated ET-1 levels promote VSMC proliferation and predispose to a procoagulant state. Aging is also associated with progressive decreases in the elastin/collagen ratio, changes in the blood vessel’s architecture and mechanical properties, and increased arterial stiffening [379,380].

Estrogen may be an appropriate option for treatment of menopausal symptoms when initiated in healthy women <60 years or within 10 years of menopause onset, but it may have plaque-destabilizing and other adverse effects in advanced atherosclerosis [381]. In a Korean nationwide study of women entering menopause at ≥40 years with no history of CVD, including MHT users and a non-MHT group, an increased risk of CVD and stroke was observed with the administration of tibolone, oral E2, or transdermal E2 compared with the non-MHT group regardless of the age of starting MHT. However, among tibolone users, a longer period from entering menopause to taking 1.25 or 2.5 mg tibolone was specifically linked to a higher risk of CVD compared with non-MHT users [319]. Also, the average age of menopause in the United States is 51 years [312]. Participants in the NHS study were aged 30–55 years, with approximately 80% starting MHT within two years of menopause [300]. In contrast, women in the HERS study were around 67 years old and had been postmenopausal for several years [377]. Also, the WHI study included older women (aged 50–79 years), with those aged 50–59 years having been menopausal for about 6 years and exhibiting subclinical atherosclerosis [293]. The “timing” hypothesis suggests that as women age, they experience subclinical vascular and myocardial changes that begin during the menopausal transition and progressively worsen over time. This “timing” or “critical window” hypothesis asserts that the timing and duration of MHT are key factors for achieving positive cardiovascular outcomes. Studies such as the Kronos Early Estrogen Prevention Study (KEEPS) and the Early versus Late Intervention Trial with Estradiol (ELITE) have evaluated how MHT timing impacts atherosclerosis progression in early menopausal women [312]. In KEEPS, women aged 42–58 years who were 6–36 months postmenopausal were given low-dose oral CEE or weekly transdermal E2, both with cyclic oral micronized P4, or a placebo for 12 days per month [312]. ELITE compared the effects of early versus late MHT initiation by randomizing women who were less than 6 years or more than 10 years postmenopausal to receive oral E2 for 2–5 years. The SMART trial, which evaluated CEE/bazedoxifene treatment for 12 months in Post-MW, found favorable changes in lipid profiles for up to 2 years, without negative effects on lipid metabolism, coagulation, hemostatic balance, cardiovascular events, or VTE incidence [382]. In healthy Post-MW, oral E2 initiated within 6 years after menopause was linked to less progression of subclinical atherosclerosis compared to placebo, whereas initiation ≥10 years after menopause did not show this benefit [315]. Also, women under 50 years who had undergone surgical menopause experienced a reduced risk of CAD when using MHT compared to non-users [67]. For women approaching menopause, MHT reduced the risk of both subclinical and clinical CAD, and the elevated risk of VTE with oral MHT formulations was lessened when using transdermal E2 [383]. Taken together, these findings underscore the importance of MHT timing, with low-dose oral E2 and transdermal E2 showing benefits when used during the 3–5 years before menopause and in the immediate postmenopausal period, and for a duration lasting for about 5 years [384,385].

### 7.6. MHT and Preexisting Conditions in Post-MW

Pre-existing cardiovascular conditions may also influence the outcomes of MHT. At baseline, participants in the HERS study had CVD [312]. In the WHI study, although the primary goal was to evaluate the benefits of MHT for primary CVD prevention in healthy women, 36% of participants had HTN, 4.4% had diabetes, 49% were smokers, 34% were obese, and 13% were undergoing treatment for hypercholesterolemia. These conditions point to an underlying atherosclerotic process that could potentially interfere with the positive effects of E2. Although some studies show no clinically meaningful effect of oral E2 on BP in women with preexisting HTN, WHI showed that CEE, either alone or in combination with medroxy-progesterone acetate (MPA), increased BP by 1 to 1.5 mmHg [386,387]. As studies have shown that a 1 mmHg reduction in systolic BP can reduce the relative risk of major CVD by 2% and reduce heart failure events by 3% [388], women with HTN should be cautious in taking MHT. The Cardiovascular Health Study found a positive association between MHT and FMD in healthy Post-MW women older than 65 years, though no effect was observed in women over 80 with CVD or cardiovascular risk factors [389]. In a study on OVX cynomolgus monkeys receiving an atherosclerotic diet, early administration of CEE resulted in a 70% reduction in coronary artery plaque size, while delayed administration after moderate atherosclerosis developed reduced plaque by only 50%, and no protective effect was observed in monkeys who had been on the atherosclerotic diet for two years before receiving CEE [390]. Also, in OVX female rats with carotid artery balloon injury, E2 treatment reduced intimal hyperplasia and VSMC proliferation when administered on the day of injury or 15 min before, continuing for three days post injury, but it had no effect when treatment was initiated seven days after injury [391]. These findings emphasize the importance of timing in MHT initiation for potentially preventing or delaying CVD progression. During the menopausal transition, women typically maintain healthy arterial function, which allows MHT to offer vascular benefits. However, as CVD progresses in Post-MW, changes such as EC senescence/dysfunction, EC-to-mesenchymal transition, and increased levels of mesenchymal cell markers, vascular permeability, VSMC migration, and proteolytic enzymes alter ECM proteins and induce vascular remodeling, leading to reduced responsiveness to E2 [392].

Sex differences and menopause-related changes in the renal, gastrointestinal, endocrine, nervous, and immune systems have also been observed, potentially influencing the development of CVD and its response to MHT [215]. For instance, diabetes increases the risk of having MI or stroke by 2-fold. The relative risk for CVD mortality in patients with diabetes is greater in women (2.42) than in men (1.86), and female diabetic patients have a worse prognosis after MI and are more likely to develop congestive heart failure [393]. Also, coronavirus disease 2019 (COVID-19) is characterized by an exaggerated immune response with hypercytokinemia, inflammatory infiltration of the lungs, and acute respiratory distress syndrome that could lead to death. The risk of severe COVID-19 outcomes is lower in women than in men worldwide, suggesting that the female biological sex confers protective mechanisms [188]. E2 and other ER modulators could influence the course of COVID-19 infection by inhibiting viral replication and its effects on the cardiovascular/pulmonary/immune system, decreasing the production of cytokines responsible for the cytokine storm, and inhibiting the interaction of IL-6 with its receptor. In this respect, E2 deficiency and MHT could have significant impact on COVID-19 prognosis in Post-MW [394].

## 8. Influence of Hormonal Environment

The vascular response to E2 could be affected by the hormonal environment, the upstream gonadotropins, the extragonadal locally produced hormones by various tissues, sex hormone-binding globulin (SHBG), and the interaction with P4 and T and their precursors.

### 8.1. MHT and Gonadotropins

Gonadotropins, including follicle-stimulating hormone (FSH) and luteinizing hormone (LH), are secreted by the pituitary gland and act on the ovaries in females to stimulate the release of E2 and P4 and on the testes in males to promote the production of T and DHT [47]. During the menopausal transition, a notable drop in circulating E2 levels is typically accompanied by a significant rise in FSH concentrations [395]. This inverse correlation between E2 and FSH indicates that E2 continues to influence pituitary FSH secretion even after menopause [396]. Findings from the KEEPS trial revealed that recently menopausal women undergoing MHT for 48 months demonstrated more pronounced associations between hormone levels in both the oral CEE and transdermal E2 groups compared to placebo. Observed variations in the relationships among E2, T, FSH, and LH across the CEE, E2, and placebo arms suggested that different MHT protocols may distinctively influence pituitary–ovarian hormone levels, feedback loops, and endocrine interactions in early Post-MW [397]. Additionally, FSH concentrations have been reported to be higher among women who had used MHT for four or more years compared to non-users [398]. Notably, a higher incidence of cardiovascular events has been linked to the use of gonadotropin-releasing hormone (GnRH) agonists compared to antagonists [399], highlighting the importance of further investigating the interplay between MHT, gonadotropin levels, and cardiovascular risk.

### 8.2. Ovarian Versus Locally Produced Sex Hormones

Previous studies have predominantly concentrated on ovarian estrogen biosynthesis, circulating E2 concentrations, and the administration of exogenous estrogens. However, investigations into estrogens synthesized locally via autocrine or paracrine mechanisms, the related enzymatic pathways, and their vascular impacts remain limited. After ovarian E2 production declines during menopause, estrogens and androgens begin to be synthesized within peripheral tissues, where they exert local effects and undergo intracellular inactivation. The distribution of enzymes responsible for steroid synthesis and deactivation is highly cell- and tissue-specific, enabling the localized generation of minimal quantities of estrogens/androgens to satisfy physiological and metabolic demands [400].

Advancing age, increased adiposity, and psychological stress have all been associated with elevated local E2 synthesis. This phenomenon is considered a compensatory response to systemic E2 deficiency during menopause and is achieved by upregulating enzymes and molecular factors that promote local E2 generation and release. In Post-MW, estrogen production from autocrine and paracrine origins is relatively enhanced [401]. Also, OVX rats continue to produce E2 for a considerable period and do not fully reach a menopausal state until 16–18 months of age. Notably, adipose tissue serves as the main site of aromatase activity, contributing to the increased peripheral conversion of androgens into E1 during the postmenopausal period [19]. Additional research has demonstrated that Post-MW show elevated aromatase mRNA levels and increased immunohistochemical signals in human aortic SMCs, with even higher levels observed in those suffering from atherosclerosis [402]. Within the hippocampus, aging in women is accompanied by enhanced expression of nuclear ERα, aromatase, and components of the Golgi apparatus, indicating the presence of locally synthesized estrogens in the brain. These estrogens may facilitate the upregulation of ERα, thereby promoting neuronal metabolism in postmenopause. Conversely, patients with Alzheimer’s disease exhibit downregulation of both canonical and alternatively spliced ERα mRNA as well as aromatase gene expression, implying a reduction in local brain estrogen production and impaired ERα signaling [403]. Consequently, the role of locally synthesized sex hormones across various tissues should be carefully evaluated when developing MHT protocols for Post-MW.

### 8.3. MHT and Sex Hormone-Binding Globulin (SHBG)

A large proportion of circulating estrogens is tightly yet reversibly bound to SHBG. This extensive binding to plasma SHBG influences the pharmacokinetics and distribution volume of estrogens, and such parameters may be altered in Post-MW who present with hepatic or renal dysfunction. In women, SHBG concentrations are regulated through a dynamic balance, with estrogens acting as stimulators and androgens, insulin, excess adiposity, and fat distribution patterns functioning as inhibitors. Although SHBG levels do not show significant differences between Pre-MW and Post-MW, Post-MW typically exhibit a pronounced decline in circulating E2 while maintaining relatively stable T levels. Additionally, SHBG shows inverse correlations with BMI, waist-to-hip ratio, insulin, and T concentrations in both Pre-MW and Post-MW, while a positive correlation with E2 is observed exclusively in Pre-MW. Regression analysis further reveals that, in Pre-MW, SHBG levels are positively associated with E2 and negatively associated with T and insulin, with no clear link to waist-to-hip ratio. In contrast, Post-MW exhibit a negative association between SHBG and waist-to-hip ratio, with no significant correlation with E2, and a weaker association with T and insulin. These findings suggest that the regulatory factors of SHBG undergo changes during the menopausal transition, and a critical threshold of E2 might play a pivotal role in determining circulating SHBG concentrations [404]. Data from a cohort of healthy Post-MW aged 41 to 92 years demonstrated that while E2 levels remained relatively stable across subgroups, SHBG levels rose progressively with age, ranging from 43 to 68 nmol/L [405]. Additionally, in women who retained their ovaries, aging was linked to a 27% reduction in androstenedione levels and a 30% increase in SHBG levels, an effect not seen in oophorectomized women [26]. Among healthy Post-MW receiving E2 therapy, elevated free E2 and SHBG levels along with decreased free T were associated with lower subclinical atherosclerosis and reduced carotid intima-media thickness, likely mediated in part by favorable changes in lipid profiles [406].

## 9. Vascular Effects of Progesterone (P4)

Progesterone (P4) serves as the primary progestogen synthesized by the female ovary and the adrenal cortex. Cytoplasmic/nuclear P4 receptors (PRs) exist in two main isoforms, PR-A and PR-B, which originate from different transcriptional start sites within the same gene [407]. PR-B is the full-length form, comprising 933 amino acids. PR-A comprises 769 amino acids and is essentially the same as PR-B but lacks the initial 164 amino acids from the N-terminus, which contain a unique AF-3. Both PR-A and PR-B are located in the cytoplasm and translocate to the nucleus upon activation, but they have different expression, distinct roles, and specific functions in different tissues. PR-B is generally a stronger activator of gene transcription than PR-A. In some cases, PR-A activation may counteract PR-B-mediated effects [408]. PR-A and PR-B isoforms have been detected in ECs and VSMCs across several species, including humans, primates, rabbits, rats, and mice [409,410]. In human VSMCs, both wild-type PRs and two splice variants, one lacking exon 4 (PRΔ4) and another lacking exon 6 (PRΔ6), have been identified. Notably, PRΔ6 acts as a dominant-negative inhibitor of the transcriptional activity of wild-type PRs in Pre-MW, but this effect is absent in Post-MW [118]. A subclass of membrane PRs (mPRα, mPRβ, mPRγ, mPRδ, and mPRε) remains bound to the plasma membrane upon activation [411]. Membrane mPRα, mPRβ, and mPRγ mediate rapid, cell-surface-initiated P4 actions in ECs and VSMCs. 

P4 exerts unique effects on the vasculature and influences the actions of E2. Like E2, studies support P4 anti-inflammatory effects through the regulation of COX, prostaglandin synthesis, and T-cell activation [190]. Also, P4 decreases innate immune inflammatory response and promotes immune tolerance and antibody production, thus mitigating the immune imbalance associated with the COVID-19 cytokine storm [188]. P4 exerts antihypertensive actions partially by modulating eNOS activity. Also, P4 promotes coronary vasodilation and NO release in pigs [412], upregulates eNOS in ovine uterine artery ECs [413], and stimulates NO-mediated relaxation in ovine uterine artery and rat aorta [414]. P4 upregulated eNOS expression in HUVECs and augmented eNOS promoter activity through a PR-A and specificity protein-1-dependent manner, while the PR antagonist RU-486 reduced eNOS expression and impaired vasodilation in rats [143]. In ECs, mPRs mediate rapid P4 signaling, resulting in increased NO synthesis [415]. Low doses of P4 could also potentiate the effects of low doses of E2 on NO production. In HUVECs, P4+E2 (5 nM) increased phosphorylation of ERK and Akt, eNOS phosphorylation, and NO production, and these effects were not observed with 5 nM of either P4 or E2 alone and were blocked with PI_3_K inhibitor wortmannin, Akt inhibitor ML-9, and MAPK inhibitors AZD6244 and U0126. Treatments with specific ER and PR agonists suggested a role of mPRα (PAQR7), GPER-1, and ERα but not ERβ in P4+E2-induced NO production. These observations suggest that a low concentration of P4+E2 rapidly increases NO production in HUVECs through mPRα, ERα, and GPER and downstream activation of PI_3_K/Akt and MAPK pathway [149]. P4 is also capable of rapidly activating COX, enhancing the production of PGI_2_, and suppressing Ang II-stimulated ET-1 release in bovine aortic ECs. In OVX SHR, P4 treatment reversed the cardiac damage caused by ovarian hormone deficiency and promoted bradykinin-induced endothelium-dependent coronary vasodilation, which was blocked by the COX inhibitor indomethacin, suggesting a role of the prostanoid pathway [148]. Low concentrations of P4+E2 also exerted genomic actions and increased mPRα, COX-1, and prostacyclin-synthase mRNA expression in HUVECs [149]. In OVX female rats, P4 treatment reversed the reduced baseline coronary perfusion pressure and restored the decreased endothelium-dependent bradykinin-induced coronary vasodilation. The P4 effects were not affected by blockade of the NO pathway or both the NO and prostanoids pathways but were abolished by combined blockade of NO, prostanoids, and epoxyeicosatrienoic acids, suggesting P4 activation of additional epoxyeicosatrienoic acids mediated hyperpolarization and vasodilation pathways in the coronary vascular bed [152]. The P4/PR-mediated coronary vasodilation in rats may be sex-specific, involving mainly NO and prostanoids in males and NO and EDHF pathways in females [416].

P4 inhibits rat aortic smooth muscle cell (RASMC) proliferation and migration [178] and facilitates the inhibitory effects of E2 on VSMC proliferation through inhibition of MAPK [266]. In pulmonary arterial SMCs, P4 reversed the IL-6-induced proliferation partly by decreasing phosphorylation of pro-proliferative Erk1/2 and Akt kinases and upregulating anti-proliferative pSmad1-Id1/2 axis. P4/PR also interacts with signal transducer and activator of transcription 3 (STAT3) and reduces IL-6-induced STAT3 nuclear translocation, chromatin binding, and transcription of downstream CCND1 and BCL2, thereby reversing IL-6-induced proliferation of pulmonary arterial SMCs. This may help explain the better prognosis of PAH in females [179]. However, other studies in RASMCs have suggested that P4 promotes cell proliferation, migration, and apoptosis; these effects need further examination [177]. P4 has also been shown to induce endothelium-independent relaxation in coronary arteries of primates, pigs, rabbits, and rats while reducing [Ca^2+^]_c_ in coronary VSMCs of both porcine and rabbit origin [161,162]. In VSMCs, activation of P4/mPRα leads to a rapid decline in [Ca^2+^]_c_ by modulating SERCA2 and phospholamban along with other signaling pathways, such as G_i_, MAPK, Akt/PI_3_K, and RhoA [167]. Additionally, P4 suppresses VSMC contraction triggered by phorbol esters and inhibits PKC activity, primarily through elevation of intracellular cAMP levels [174].

However, P4 produces less vasorelaxation than E2 and antagonizes the favorable effects of E2 on NO production and vascular relaxation in the canine and porcine coronary artery [417]. Also, P4 counteracts the antioxidant effect of E2 by enhancing NADPH oxidase activity and ROS production and down-regulating superoxide dismutase in OVX mice [154]. In mouse renal arterial ECs, P4 increased ROS levels and abolished the H_2_O_2_-induced antioxidant response, glutathione production, and glutathione peroxidase activity [418]. P4/PR interaction with EC mineralocorticoid receptor (MR) could also contribute to the enhanced obesity-associated leptin-induced EC dysfunction in females versus males. In support, genetic deletion of either endothelial MR or PR in female mice prevents leptin-induced endothelial dysfunction [419]. P4 also upregulates vascular AT_1_R, promotes vasoconstriction [420], and diminishes E2 anti-inflammatory benefits in ischemic brain injury [15].

P4 could modify the effects of E2 on women’s health and the cardiovascular system. In women who have an intact uterus, P4 is co-administered with E2 to mitigate the risk of endometrial cancer [421]. Oral micronized P4 has also been shown to alleviate menopausal symptoms and enhance sleep quality [422,423]. Furthermore, the use of vaginal micronized P4 does not affect the metabolic and cardiovascular outcomes associated with non-oral E2 in early Post-MW [424]. Other studies showed that micronized P4 in combined MHT had a neutral effect on VTE, MI, and ischemic stroke [425]. In the REPLENISH trial, treatment of vasomotor symptoms in Post-MW with TX-001HR (E2+P4) did not affect glucose levels, total cholesterol, triglycerides, coagulation factors, or VTE incidence [426]. Other studies showed no attenuation of CEE-induced vasodilation by micronized P4 or medroxy-progesterone acetate (MPA) [427]. Also, the NHS showed a similar risk reduction for CAD in women taking CEE alone or CEE+MPA [300]. In rabbits, CEE and E2 conferred anti-atherosclerotic actions that were not prevented by MPA or other progestins [293].

Drospirenone (DRSP), a 17α-spironolactone derivative, binds to PR with an affinity comparable to that of natural P4. DRSP is commonly combined with ethinyl-E2 for use as an oral contraceptive and in MHT. It exerts stronger anti-androgenic effects than P4 and has anti-mineralocorticoid properties that oppose the salt-retaining effects of E2. Angeliq (E2+DRSP) reduces vertigo/dizziness and carotid intima-media thickness [428] and improves vascular parameters, EC function, reactive hyperemia, and vascular protection in normotensive early Post-MW [429]. Also, E2+DRSP, in combination with an angiotensin-converting enzyme inhibitor (ACEI) and angiotensin receptor blocker (ARB), lowers BP in hypertensive Post-MW [430]. In Post-MW with metabolic syndrome, both E2+DRSP and E2+dydrogesterone lower fasting blood glucose levels, but only E2+DRSP is effective in improving glycemic excursions, insulin sensitivity, lipid profiles, and inflammatory markers [431].

Other studies suggest that some progestins may negatively impact the vascular effects of MHT and diminish the beneficial effects of E2 on lipid profiles, and they could counteract the protective effects of CEE on coronary atherosclerosis and reduce E2-induced NO release in Post-MW. In healthy Post-MW, resting vascular resistance and resistance after cold pressor stimulation remained unchanged after 21 days of E2 therapy alone but significantly increased when MPA was added [432]. When Post-MW were treated with E2 for two weeks, followed by E2+norethisterone acetate for another two weeks, cGMP excretion was elevated in both phases, but PGI_2_ metabolites were higher in the E2 phase, whereas TXA_2_ metabolites increased in the E2+norethisterone acetate phase [433]. Also, oral E2/MPA increases TXA_2_ metabolite, but transdermal E2/MPA does not [434]. Additionally, treatment of human female coronary ECs with MPA or norethisterone acetate enhanced E2-induced reduction in MMP-1 and in turn affected plaque stability. Furthermore, CEE alone or CEE+MPA increased MMP-9 levels in Post-MW with established CAD [421]. In the HERS trial and one arm of the WHI trial, the heightened stroke risk in women taking CEE+MPA compared to non-users of MHT was related to MPA. Additionally, in the Postmenopausal Estrogen/Progestin Interventions (PEPI) trial, MPA reduced the positive effects of CEE on LDL-C and HDL-C levels [293]. In cynomolgus monkeys, E2 or E2+P4 exhibited similar anti-atherosclerotic effects, which were not observed in monkeys treated with CEE+MPA [435]. Notably, in surgically postmenopausal monkeys on an atherogenic diet, lipid lowering improved ACh-induced vasodilation, and CEE had no impact on vascular reactivity, but MPA impaired the beneficial effects of CEE on coronary flow reserve [436]. Also, while E2 alone improved cardiovascular function in OVX diabetic rats by reducing inflammatory cytokines and improving metabolic parameters, P4 alone or E2+P4 did not show substantial benefits [437]. Additionally, MPA negated the beneficial effects of E2 in reducing balloon injury and intimal thickening in rat models [438]. These complex interactions between E2 and P4 on vascular function could determine the outcome of combined E2/P4 therapy in postmenopausal CVD. Specifically, the impact of MPA on the absence of vascular benefits from MHT in RCTs requires further investigation.

## 10. Vascular Effects of Testosterone (T)

Androgens play a crucial role in maintaining reproductive function, secondary sexual characteristics, libido, metabolic processes, muscle mass, bone structure, and brain function in both men and women. Most of the androgens are produced by the testes, adrenal glands, and ovaries. Typically, about 10% of circulating T is irreversibly converted into its more potent derivative DHT by the enzyme 5α-reductase, found in tissues like the liver, prostate, genital skin, and hair follicles. T is also aromatized to E1 or E2, primarily in adipose tissue, which contributes to its indirect cardiovascular effects. This is supported by the observation that aromatase inhibitors reduce EC vasodilator function, likely by blocking T conversion to E2.

During the reproductive years, women produce 25% of circulating T from the ovaries, another 25% from the adrenal glands, and the remaining 50% from the peripheral conversion of androstenedione. Although serum T levels in women are significantly lower than in men, these levels do not substantially decrease after menopause, as the ovaries continue to produce androgens [28]. Bilateral oophorectomy in Pre-MW causes a reduction of around 50% in circulating T concentrations [17]. Also, women who undergo surgical bilateral oophorectomy have T levels 40–50% lower than those experiencing natural menopause, suggesting that natural menopause is a relatively hyperandrogenic state [439]. Longitudinal studies in Post-MW over 10 years have shown an age-related increase in total serum T and androstenedione alongside a decrease in E2 and DHT [440]. During the postmenopausal period, total T levels gradually increase with age from 19.9 to 26.2 ng/dL, whereas free T levels remain relatively stable between 3.7 and 4.6 pg/mL [405]. However, circulating levels of E2 and T may not accurately reflect their tissue concentrations, as both aromatase and 5α-reductase are present in various tissues, including blood vessels [441]. In the postmenopausal stage, each cell and peripheral tissue alongside the adipose tissue synthesizes small amounts of androgens to meet local metabolic needs [400]. Consistent with this concept, BMI has been linked to androgen metabolism in Post-MW [442].

Androgens bind to androgen receptors (ARs) in various tissues, including the cardiovascular system. The sensitivity of tissues to androgens depends on the activity of androgen metabolic enzymes, AR levels, polymorphisms, and co-regulators [443]. The use of aromatase and 5α-reductase inhibitors, often in breast cancer treatments, can alter androgen levels and lead to cardiovascular effects. Androgens might influence cardiovascular risk in Post-MW, with the relationships between circulating free E2, free T, SHBG, and their ratios being more predictive of carotid intimal thickening and the progression of CVD than the individual hormone levels alone. In Post-MW, higher E2 levels were associated with a reduced risk of CAD, whereas higher T levels and a higher T/E2 ratio were linked to an increased risk of cardiovascular events, CAD, and heart failure [55]. In primate VSMCs, E2+T upregulates AR expression, while E2 alone has minimal effects, suggesting an interaction between E2 and T in regulating ARs [444]. ARs are also expressed in rat ECs and VSMCs although at lower levels in females compared to males [445].

Although elevated T levels are linked to lower all-cause and cancer mortality in men, they may negatively affect cardiovascular and renal health in both men and Post-MW [446]. The higher BP and incidence of CVD in adult men are partially attributed to androgen-induced harmful cardiovascular effects. Supporting this concept, T impairs bradykinin-induced endothelium-dependent relaxation and NO production in porcine coronary arteries while enhancing TXA_2_-induced coronary vasoconstriction in guinea pigs [146]. Additionally, the AR antagonist flutamide induces vasodilation in rat blood vessels [447]. The increased cardiovascular and renal disease risk observed in Post-MW has also been related to a higher T/E2 ratio due to decreased ovarian E2 production. Furthermore, BP is higher in male compared to female SHR, decreases in castrated males, and rises in OVX females treated with T, indicating a connection between the increased T/E2 ratio and HTN after menopause [439]. Also, male mice fed a high-fat diet show reduced ACh-induced relaxation and increased ROS generation in aortic rings, and these effects are reversed in castrated mice and restored in mice treated with T, suggesting that T promotes oxidative stress and EC dysfunction during a high-fat diet [155]. In cardiomyocytes, DHT shows stronger binding to ARs than T and induces hypertrophy through activation of mTOR complex 1/40S ribosomal protein S6 kinase 1 [200]; upregulation of Ca^2+^ regulatory proteins, IP_3_/Ca^2+^, and MAPK; and increased contraction mechanisms [192]. In female mice, DHT treatment led to a more than twofold increase in carotid arterial stiffness and a reduction in aortic expression of ERα and GPER. Also, in cultured VSMCs, treatment with DHT, dexamethasone, or MPA resulted in a suppression of ERα expression, while DHT specifically inhibited GPER mRNA, indicating that androgens may contribute to arterial stiffening by downregulating ER expression [448].

Other research points to cardiovascular benefits of T and suggests that it influences lipid metabolism, enhances hepatic lipoprotein lipase activity, and reduces serum triglycerides and atherogenic lipids [165,449,450]. Lower serum T levels have been observed in men with chronic CVD compared to healthy men, and low circulating T levels are linked to increased atherosclerosis, CAD, and cardiovascular events [450,451], implying that T could affect the progression of angina, myocardial ischemia, and heart failure. Epidemiologic studies have also shown low serum T levels in men with abdominal aortic aneurysm (AAA). Furthermore, androgen deprivation therapy may exacerbate cardiovascular complications in prostate cancer patients. Androgen synthesis inhibitors are also associated with higher incidences of HTN, atrial tachyarrhythmia, and heart failure [399]. T might support coronary vasodilation in young men, while the elevated risk for angina and MI in older men could be related to reduced T levels. Notably, in men with established CVD, intracoronary T administration has been shown to increase coronary blood flow and alleviate myocardial ischemia [452]. Studies have also indicated that women with lower T levels have a higher incidence of cardiovascular events compared to those with higher T levels [453]. In women aged 50–59 years, T shows a positive association with low LDL-C and high HDL-C levels, while women with CVD tend to have lower T levels [454]. In healthy Post-MW, there is a positive correlation between plasma T levels and brachial artery FMD, implying that T may offer protective effects on ECs [455]. Elevated free T levels are also linked to a reduced risk of heart failure in Post-MW [74].

Experimental research further supports the vascular benefits of T. T treatment was shown to protect against vascular aging, improve cell senescence, and reduce intima-media thickness and vascular remodeling in the carotid artery and abdominal aorta of aging mice through the Gas6/Axl pathway [456]. Also, in a male murine model of AAA induced by topical CaCl_2_ application and Ang II-infusion, AAA formation, aortic inflammation, macrophages infiltration, and IL-6 expression were exacerbated in castrated mice and flutamide-implanted intact mice and prevented in T-treated mice, suggesting that selective AR modulators might be useful in aortic pathologies [457]. Also, DHT stimulates EC proliferation and angiogenesis by upregulating VEGF mRNA expression [159]. Additionally, T has been shown to induce relaxation in the coronary arteries of canines, pigs, and rabbits as well as in the aorta of rabbits and rats [161,458,459]. Canine intracoronary administration of T induces vasodilation and NO release [460]. In human aortic ECs, physiological T concentrations (1–100 nM) induced rapid increases in NO production and eNOS phosphorylation/activation that were simulated by DHT and blocked by the AR antagonist nilutamide, transfection with AR siRNA, Akt inhibitor SH-5, or PI_3_K inhibitor wortmannin, supporting that T rapidly induces EC NO production via AR-dependent and PI_3_K/Akt-mediated phosphorylation/activation of eNOS [144]. In human aortic ECs, T rapidly assembles a membrane complex of AR, caveolin-1, and c-Src, thereby facilitating a c-Src/PI_3_K/Akt cascade and subsequent activation of eNOS [145]. The vascular relaxation effect induced by T is partially diminished in the presence of K^+^ channel blockers, indicating the involvement of voltage-dependent K^+^ channels [458]. In human coronary artery ECs, T, non-permeant T conjugate, or the non-aromatizable DHT caused hyperpolarization and rapidly enhanced small- and large-conductance Ca^2+^-activated K^+^ currents (SK_Ca_ and BK_Ca_), which were blocked by the AR antagonist flutamide, pertussis toxin, protein kinase A inhibitor H-89, or SK_Ca_ and BK_Ca_ channel blockers apamin and iberiotoxin, supporting T-induced rapid activation of SK_Ca_ and BK_Ca_ currents via a membrane AR, G_i/o_ protein, and protein kinase A [151]. In the aorta of SHR, T facilitates the release of EDHF and activates both voltage-dependent and BK_Ca_ channels. Nonetheless, a considerable portion of T-mediated vasodilation in both WKY and SHR occurs independently of the endothelium and is attributed to activation of ATP-sensitive K^+^ channels in VSMCs [461]. DHT influences the proliferation of human umbilical VSMCs in a concentration-dependent fashion, enhancing [^3^H]thymidine incorporation at low doses and suppressing it at higher doses [180]. Also, T promotes apoptosis in VSMCs through mitochondrial ROS generation [181]. Both T and DHT cause relaxation of the porcine coronary artery and a reduction in VSMC [Ca^2+^]_c_ [161,162,458]. In porcine coronary artery, T induces more prominent inhibition of PGF_2α_- than KCl-induced contraction [161], implying inhibition of other PGF_2α_-activated contraction pathways, such as PKC. In cultured VSMCs, T ameliorated Ang II-induced cell senescence and collagen overexpression, decreased MMP-2 expression/activity, and increased the expression of TIMP-2, underscoring the protective effects of T on VSMCs senescence, collagen synthesis, and vascular aging [462].

Conditions linked to hormone imbalances may offer valuable insights into how the hormonal environment affects cardiovascular function. Polycystic ovary syndrome (PCOS) is marked by polycystic ovaries, amenorrhea, hirsutism, anovulatory infertility, elevated androgen and estrogen levels, obesity, insulin resistance, and lipid metabolism abnormalities. In individuals with PCOS, both insulin and LH promote androgen synthesis in ovarian theca cells, leading to higher levels of T and androstenedione. Hyperandrogenemia associated with PCOS may also contribute to BP dysregulation and a heightened cardiovascular risk through direct effects of T on the vascular system, baroreflex-mediated BP control, and renal responses to baroreceptor unloading [463]. Additionally, as many women with PCOS are overweight, the increased androgen levels, along with their conversion by aromatase, may elevate E1 concentrations in adipose tissue. Some research in PCOS patients has suggested a link between reduced ovarian E2 and CVD [72]. While women with PCOS exhibit CVD risk factors such as elevated BP, BMI, fasting glucose, hypertriglyceridemia, and low HDL-C, a retrospective study conducted in the UK did not find a higher incidence of cardiovascular morbidity or mortality among PCOS women [464]. Notably, Post-MW women and those with PCOS share similar sex hormone profiles characterized by E1 as the predominant estrogen, a deficiency of P4, and a relative rise in androgens. Additional research in women with PCOS and animal models could offer valuable insights into how other sex hormones influence the vascular effects of E2.

In men and Post-MW, exogenous androgens are sometimes administered off-label to improve libido, sexual health, and physical performance [465]. Some studies suggest that misuse of androgenic-anabolic steroids may increase the risk of CVD in males. Nonetheless, managing hypogonadism in older men has yielded inconsistent outcomes with respect to cardiovascular health [466]. In the T Cardiovascular Trial in older men, T treatment did not lead to a higher incidence of cardiovascular or prostate-related adverse events compared to placebo, although it was linked to an increase in the volume of noncalcified coronary artery plaques [467]. Also, the TRAVERSE (TheRapy for Assessment of long-term Vascular events and Efficacy ResponSE) study in hypogonadal men 45 to 80 years old with pre-existing or a high risk of CVD showed that daily transdermal 1.62% T gel did not increase the risk for major adverse cardiac events or prostate cancer [468]. However, in male Wistar rats, T treatment was associated with increased collagen deposition in the myocardial tissue, likely through downregulation of dickkopf1 (DKK1) mRNA expression, which may explain some of the cardiac structural abnormalities observed in some androgens abusers [469]. As a result, the Food and Drug Administration introduced stricter regulations on T replacement therapy and called for additional research to evaluate its cardiovascular safety [466]. Studies on the relationship between T and CVD events in Post-MW have yielded conflicting results, demonstrating that both high and low T levels are associated with an increased risk of CVD. The only evidence-based indication for T therapy is for the treatment of hypoactive sexual desire disorder in Post-MW at doses that approximate physiological T levels in Pre-MW [470]. Because oral T causes more side effects and hepatotoxicity and requires administration more than once a day, transdermal preparations are preferred [471].

Transgender and nonbinary individuals often receive gender-affirming hormone therapy to align their outward appearance with their gender. However, the renal and metabolic processes, parameters, and biomarkers are different in transgender versus cisgender individuals [472]. Extended exposure to E2, such as through oral contraceptives or gender-affirming hormone therapy, may contribute to altered CVD risk in both cisgender and transgender females. Research utilizing human arterioles obtained from discarded adipose tissue during surgery has demonstrated that NO-dependent FMD is generally preserved in females across different age groups. However, microvascular exposure to E2 for 16–20 h led to EC dysfunction in arterioles from both younger (<40 years) and older (≥40 years) females, with even more pronounced EC and VSMC impairment observed in vessels from biological males. Notably, females aged ≥40 and biological males exhibited reduced expression of ERβ and GPER in ECs when compared to females under 40 years old. Furthermore, ERα, the receptor most closely associated with the beneficial effects of E2, was found exclusively in the adventitia. These findings have suggested a potential cardiovascular risk linked to prolonged hormone exposure via contraceptive use or gender-affirming treatments [255]. This aligns with reports that the incidences of VTE and stroke are higher in transwomen receiving gender-affirming E2-based therapy than in cisgender men, which could be related to prolonged use and potential prothrombotic effects of E2 [473,474,475]. Also, some studies have suggested that transwomen receiving transgender E2 therapy have a higher incidence of MI than cismen [474]. However, other studies showed less pronounced disparities in MI incidence rates between transwomen and cismen [476], suggesting that factors beyond hormone therapy, such as socio-economic status and stress, may be involved. The impact of T on cardiovascular function, renal health, and metabolic conditions may vary significantly between transgender men (female to male) and cisgender women. Some studies showed a higher incidence of MI in transmen receiving gender-affirming T-based therapy than ciswomen [473]. These findings underscore the importance of tailored cardiovascular risk management for each individual and reveal the gaps in current research regarding the effects of various hormone therapies and regimens in different genders [473,476]. With larger studies, expanded cohort size, and extended follow-up in the transgender and nonbinary community, the data will allow further analyses of critical health endpoints across various categories of CVD and their potential modulation by different hormone formulations, routes of administration, and doses [475,476].

## 11. Discussion and Perspectives

Sex-based differences are recognized not only within the reproductive system but also across the cardiovascular pathophysiology, contributing to variations in CVD incidence between males and females as well as between Post-MW and Pre-MW. These disparities are influenced by gene expression associated with the X and Y chromosomes, along with age-dependent changes in the levels and balance of circulating E2, P4, and T. A decline in E2 levels is a hallmark of menopause, leading to characteristic menopausal symptoms and correlating with increased CVD risk. MHT helps alleviate menopausal symptoms by restoring E2 concentrations. Both epidemiological evidence and experimental research point toward vascular protective effects mediated by E2. CVD prevalence tends to be higher in adult males and in Post-MW compared to Pre-MW. Earlier observational studies proposed a protective vascular role for MHT. Also, mechanistic investigations have uncovered sex-specific variations in cardiovascular responses and revealed beneficial actions of E2/ERs, including promotion of vasodilation and inhibition of VSMC proliferation through both genomic and nongenomic pathways. Despite these findings, RCTs have not endorsed MHT for the primary prevention of CVD and have instead highlighted potential adverse effects, such as heightened risks of VTE and stroke.

Optimizing MHT by carefully selecting its type, dosage, administration route, timing, and duration alongside evaluating age-related alterations in ER subtypes, their distribution, responsiveness, and downstream signaling may enhance its effectiveness in alleviating menopausal vasomotor symptoms and improving vascular outcomes in CVD (Table 4). With the development of newer MHT formulations and optimizing factors such as dosage, route of delivery, and timing in relation to the menopausal stage, the cardiovascular benefits of MHT can be amplified in Post-MW. Reassessment of data from older RCTs including WHI by participant age and time since menopause suggests that for healthy women within 10 years of menopausal transition who have bothersome menopausal symptoms, the benefits of MHT outweigh its risks, with fewer CVD events in younger versus older women [477]. Future research should explore the hormonal milieu and age-associated shifts in endogenous levels of E2, FSH, SHBG, P4, and T, along with their metabolites, interactions, and potential links to CVD risk, especially during the menopausal transition. Additionally, sex hormones influence lipid profiles and coagulation pathways and play regulatory roles in the gut microbiota, immune function, and nervous system, all of which may indirectly impact CVD and its response to MHT.

Current guidelines suggest increased risk for endometrial cancer among women with a uterus taking E2-only MHT. Also, MHT is not indicated for primary or secondary prevention of CVD, dementia, or deterioration of cognitive function because of the greater risks of CAD, stroke, and VTE [477,478]. Despite this, MHT has proven effective in relieving vasomotor symptoms associated with menopause. A more detailed investigation into the cardiovascular impact of sex hormones across various formulations and treatment regimens as well as the vascular signaling alterations during menopause could refine the MHT strategies and optimize their efficacy in lowering the risk and the management of CVD in Post-MW.

We should note some of the limitations of this review. Our inclusion criteria may have inadvertently excluded relevant studies, particularly those published in non-English language or in less accessible journals, and some pertinent information may have been missed. Also, while our intended focus was on MHT and CVD in Post-MW, a substantial amount of information was derived from studies in men or male animals. Future reviews could address these gaps by employing broader language inclusion and describing more large-scale comprehensive studies in women and detailed experiments in female animals as they become available.

## 12. Conclusions

Understanding the specific effects of sex hormones and how different hormonal therapies affect various biological processes in a gender-specific manner is essential for effective CVD management. Investigating the distinct roles of E2, P4, and T as well as their interactions and metabolites could provide deeper insights into personalized and effective treatment strategies. This comprehensive knowledge would support the development of targeted therapies that optimize benefits while minimizing risks, ultimately contributing to improved cardiovascular outcomes across diverse populations. Rigorous research into these dynamic interactions is crucial to refine hormone therapy practices and tailor interventions to the unique physiological needs of each individual.

## Figures and Tables

**Figure 1 ijms-26-05078-f001:**
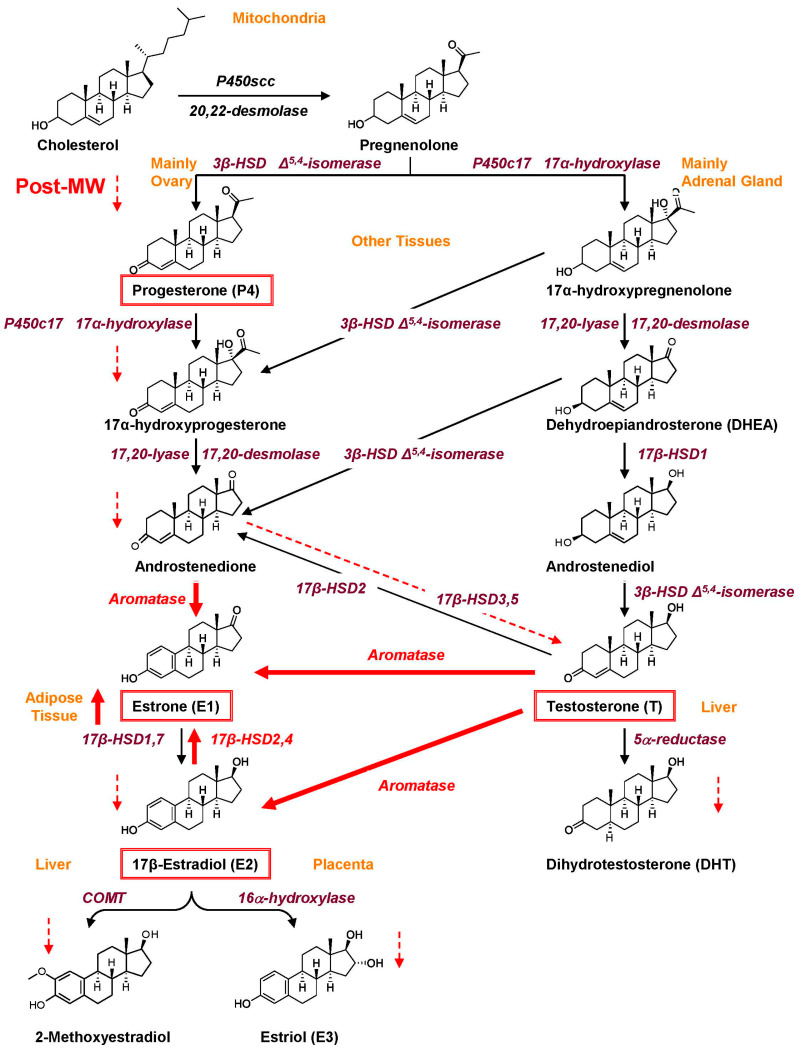
Changes in sex hormone synthesis in the postmenopausal period. In Pre-MW, cholesterol is transported into the mitochondria by steroidogenic acute regulatory protein and then converted by P450Scc (20,22-desmolase) into Δ^5^-3β-hydroxysteroid pregnenolone. In the ovary, pregnenolone is released into the cytosol, where it undergoes oxidation and isomerization by 3β-hydroxysteroid dehydrogenase (3β-HSD, Δ^5^-Δ^4^ isomerase) into the ketosteroid progesterone (P4). P4 is converted mainly into androstenedione then estrone (E1). Androstenedione could also be transformed via 17β-HSD3,5 into testosterone (T), which is then converted by aromatase into E1. E1 is transformed by 17β-HSD1,7 into 17β-estradiol (E2) and reversibly by 17β-HSD2,4 mainly in adipose tissue from E2 to E1. E2 is metabolized mainly in the liver by catechol-o-methyl transferase (COMT) into 2-methoxy-E2. During pregnancy, E2 is metabolized in the placenta by 16α-hydroxylase into estriol (E3). Pregnenolone is also converted in the adrenal glands into androstenediol, which is then converted by 3β-HSD Δ^5,4^ isomerase into testosterone (T). T is transformed in the liver by 5α-reductase into dihydrotestosterone (DHT). Sex hormones are also synthesized de novo or through interconversion by aromatase and other enzymes in the uterus, breast, and other tissues so that each cell can make enough hormones for its metabolic needs. In Post-MW, a sharp decline in ovarian function leads to decreases in the release of P4, E2, and ovarian T precursors, but the adrenal glands continue to produce T, leading to a small, slow decline in circulating free T and relative increase in T/E2 ratio. Also, in Post-MW, ovarian production and circulating levels of both E2 and E1 are reduced, but aromatase and 17β-HSD2,4 activities are increased in adipose tissue, causing relative increases in the E1/E2 ratio. Dashed red lines indicate decrease. Thick red lines indicate increase.

**Figure 2 ijms-26-05078-f002:**
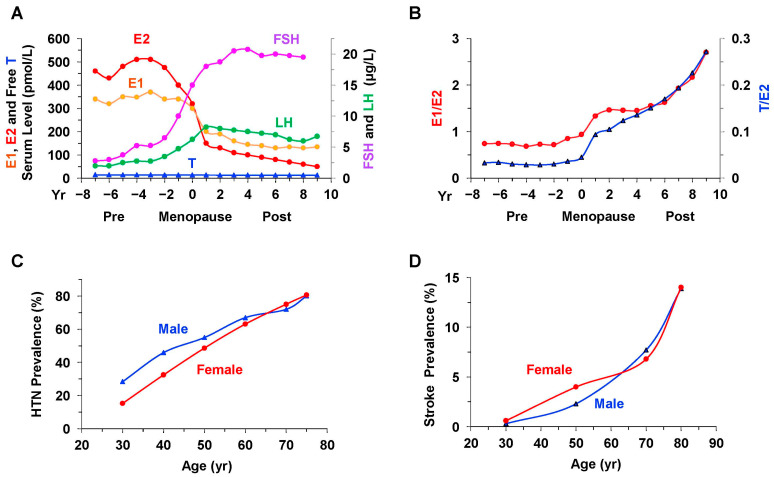
Based on previously published reports, predictive estimates of changes in sex hormone levels and the incidence of CVD during menopause have been drawn. Throughout the menopausal transition, E2 levels experience a sharp decline, and E1 levels decrease more gradually, while FSH and LH levels rise, and free T levels decline slowly and modestly (**A**), but E1/E2 and T/E2 ratios are increased (**B**) [27,28]. Changes in sex hormone levels may contribute to the observed sex differences in the incidence of CVD. Prior to menopause, women experience a significantly lower incidence of HTN compared to men. However, as ovarian function declines during menopause and postmenopause, the prevalence of HTN increases, eventually matching and even surpassing that of men in older Post-MW (**C**) [1]. Women between 25 and 45 years old experience a higher incidence of stroke compared to men, and this trend persists after menopause. However, as age increases, the gender gap gradually narrows, with men eventually matching or surpassing women at very advanced ages (**D**) [1].

**Figure 3 ijms-26-05078-f003:**
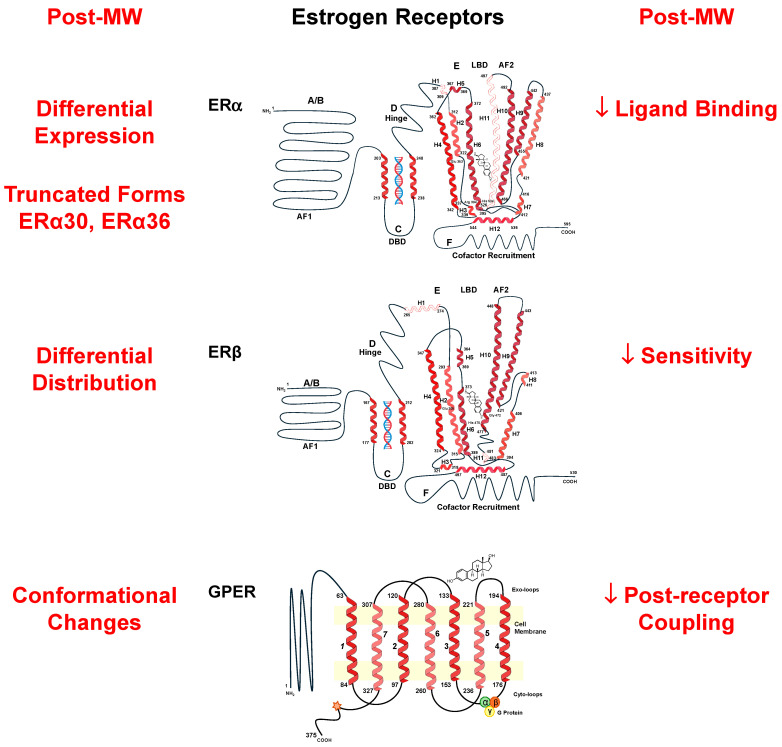
Changes in estrogen receptors (ERs) postmenopause. In Pre-MW, E2 binds to cytosolic/nuclear ERα and ERβ and membrane GPER to generate cardiovascular benefits. In Post-MW, ERs show differential expression of subtypes and variants, altered distribution, conformation changes, and decreased ligand binding, receptor sensitivity, and post-ER coupling mechanisms, leading to decreased cardiovascular benefits. ↓, decreased.

**Figure 4 ijms-26-05078-f004:**
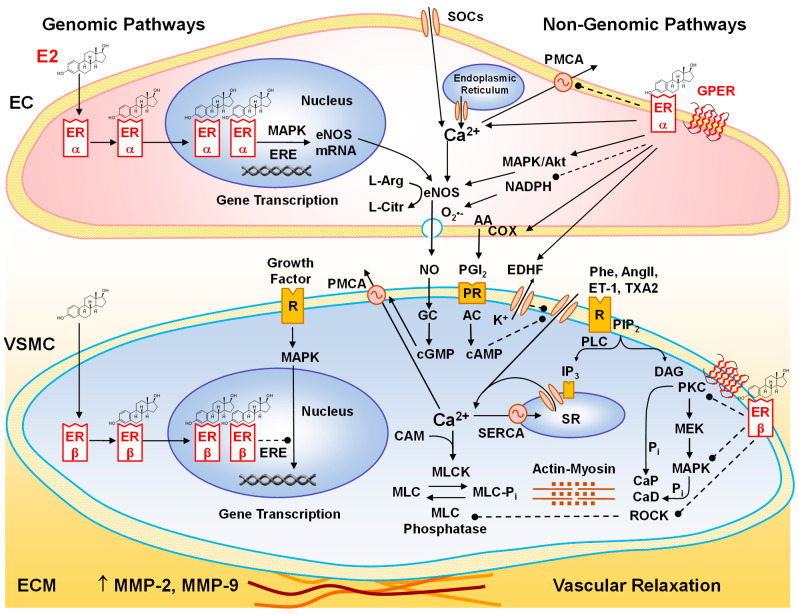
Genomic and nongenomic estrogen (E2)/estrogen receptor (ER)-mediated pathways in endothelial cells (ECs), vascular smooth muscle cells (VSMC), and extracellular matrix (ECM). In ECs, E2 binds to cytosolic/nuclear ERα and ERβ to form ER homodimers or heterodimers and stimulate genomic pathways involving MAPK activation, gene transcription, EC growth, and eNOS expression. E2 also binds less abundant plasmalemmal ERα, ERβ, and GPER in ECs and activates phospholipase C (PLC) to produce inositol 1,4,5-triphosphate (IP_3_) and diacylglycerol (DAG). E2/ER activation increases cytosolic free Ca^2+^ concentration ([Ca^2+^]_c_) through initial IP_3_-induced Ca^2+^ release from the endoplasmic reticulum and sustained store-operated Ca^2+^ entry through store-operated channels (SOCs). E2 also inhibits plasma membrane Ca^2+^-ATPase (PMCA) activity in ECs, thus reducing Ca^2+^ extrusion and contributing to the prolongation of Ca^2+^ signal. The E2-induced increase in Ca^2+^ binds to calmodulin (CAM), and the Ca^2+^/CAM complex initiates translocation of eNOS away from caveolin. E2 also activates phosphatidylinositol 3-kinase (PI_3_K) to transform phosphatidylinositol-4,5-bisphosphate (PIP_2_) into phosphatidylinositol 3,4,5-trisphosphate (PIP3) and activate Akt. ER-mediated activation of MAPK/Akt causes phosphorylation and full activation of eNOS, which transforms L-arginine to L-citrulline and produces NO. NO diffuses into VSMC and activates guanylate cyclase (GC) to form cyclic guanosine monophosphate (cGMP), which activates protein kinase G (PKG). PKG decreases [Ca^2+^]_c_ by stimulating plasmalemmal Ca^2+^ extrusion pump (PMCA) and sarcoplasmic reticulum (SR) Ca^2+^ uptake pump (SERCA). PKG also phosphorylates and inhibits myosin light-chain kinase (MLCK) and in turn decreases the actin–myosin myofilaments force sensitivity to [Ca^2+^]_c_, leading to VSM relaxation. E2 also inhibits NADPH and formation of O_2_^•−^ and ONOO^−^, thus promoting antioxidant effects that increase NO bioavailability. E2 also activates cyclooxygenases (COX) to produce prostacyclin (PGI_2_), which binds the prostanoid receptor (PR) in VSMC and activates adenylate cyclase (AC) to form cyclic adenosine monophosphate (cAMP), which activates protein kinase A (PKA). PKA decreases [Ca^2+^]_c_ and myofilament force sensitivity to Ca^2+^ using the same mechanisms as cGMP/PKG, leading to VSM relaxation. E2 also increases endothelium-derived hyperpolarizing factor (EDHF), causing activation of VSM K^+^ channels, hyperpolarization, inhibition of Ca^2+^ influx through Ca^2+^ channels, and VSM relaxation. In VSMCs, E2 binds cytosolic/nuclear ERα and ERβ to activate genomic pathways and inhibit MAPK, gene transcription, and VSMC proliferation. Also, in VSMCs, vasoconstrictor agonists such as phenylephrine (Phe), ET-1, TXA_2_, or Ang II activate their specific receptors (R) to stimulate PLC and generate IP_3_ and DAG. IP_3_ stimulates Ca^2+^ release from the sarcoplasmic reticulum (SR). Agonists also stimulate Ca^2+^ entry through Ca^2+^ channels. Ca^2+^ binds CAM to activate myosin light-chain kinase (MLCK), causing MLC phosphorylation, actin–myosin interaction, and VSM contraction. DAG activates PKC to phosphorylate calponin (CaP) or activates MAPK kinase (MEK) and MAPK cascade, leading to phosphorylation of caldesmon (CaD) and increased myofilament sensitivity to [Ca^2+^]_c_. E2 binds less abundant plasmalemmal ERα, ERβ, and GPER to activate non-genomic pathways and inhibit agonist-activated mechanisms of VSM contraction, including Ca^2+^ channels, PKC, MAPK, and ROCK-mediated [Ca^2+^]_c_ sensitization pathways. E2/ER also increases the proteolytic activity of MMP-2 and MMP-9, leading to breakdown of ECM proteins, changes in elastin/collagen ratio, and vascular remodeling. Dashed lines indicate inhibition. ↑, increase.

**Figure 5 ijms-26-05078-f005:**
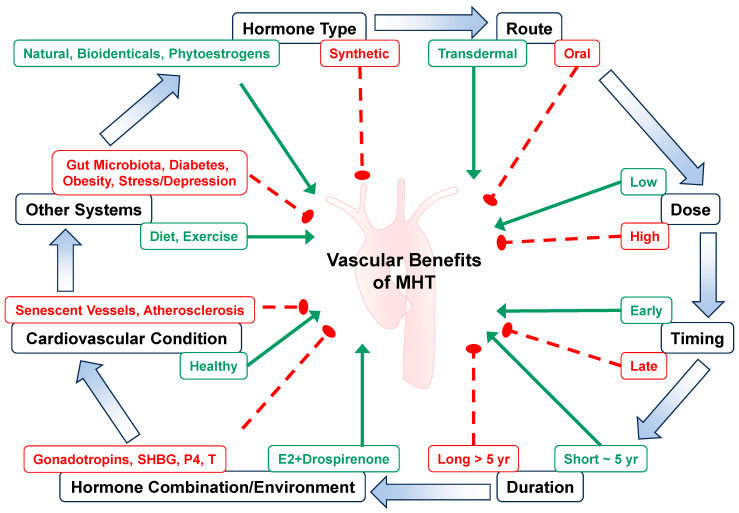
Factors contributing to the limited efficacy of MHT in managing CVD include inappropriate selection of MHT formulation, administration route, dosage, initiation timing, treatment duration, hormonal milieu, as well as the presence of underlying cardiovascular or other medical conditions (indicated in red). To enhance the cardiovascular benefits and overall effectiveness of MHT in reducing CVD risk, the use of natural estrogens, bioidentical hormones, or phytoestrogens, preferably via parenteral administration at lower doses initiated early in the menopausal transition for a limited period with an optimal hormone combination and in Post-MW with intact vascular health and no preexisting cardiovascular or systemic disorders (indicated in green) could be more advantageous.

**Table 1 ijms-26-05078-t001:** Effects of sex hormones on cardiovascular cells and biological processes.

Cell/Process	E2	P4	T	References
ECs				
Endothelium-dependent vascular relaxation	+	+	+/-	[140,141,142,143,144,145,146]
eNOS, NO	+	+	+/-	[140,141,142,143,144,145,146]
PGI_2_	+	+		[147,148,149]
EDHF	+	+	+	[150,151,152]
ROS production	-	+	+	[153,154,155]
Proliferation	+		+	[156,157,158,159]
Migration	+			[156,157,158]
Angiogenesis	+			[158,160]
VSMCs				
Endothelium-independent vascular relaxation	+	+	+	[87,161,162]
[Ca^2+^]_c_, L-type Ca^2+^ channel	-	-	-	[161,162,163,164,165,166]
PMCA, SERCA	+	+		[167,168]
BK_Ca_	+		+	[169,170,171]
K_v_, K_ATP_			+	[169,172]
PKC	-	-		[173,174]
ROCK	-			[175]
Proliferation	-	+/-	+/-	[100,176,177,178,179,180]
Migration	-	+/-		[100,176,177,178]
Apoptosis		+	+	[177,181]
Fibroblasts				
MMP-2, MMP-9	+/-			[182,183,184]
Fibrosis	-			[182,185,186]
Inflammation	-	-	+/-	[112,187,188,189,190]
Atherogenesis	-		-	[15,165]
Cardiomyocytes				
β-adrenergic receptor			+	[191,192]
L-type Ca^2+^ channel	-		+	[191,192,193]
Delayed-rectifier K^+^ channel	+			[193]
SERCA2a	+	+		[194,195]
K_v_2.1 channel		+		[196]
Contractility	-	-	+	[191,192,193,197]
Hypertrophy	-		+	[100,198,199,200,201]
Glucose uptake			+	[202]
ATP synthesis	+			[203]
ROS Production	-		+	[203,204,205]
β-oxidation, cell respiration		+		[206]
Proliferation		+		[47,207]
Apoptosis	-			[47,208]

BK_Ca_, large-conductance Ca^2+^-activated K^+^ channel; K_ATP_, ATP-sensitive K^+^ channel; K_v_, voltage-sensitive K^+^ channel; PKC, protein kinase C; ROCK, rho-kinase; PMCA, plasma membrane Ca^2+^-ATPase; SERCA, sarcoplasmic/endoplasmic reticulum Ca^2+^-ATPase. +, stimulate; -, inhibit.

**Table 2 ijms-26-05078-t002:** Outcomes of representative MHT clinical trials in CVD (in chronological order).

Clinical Trial	Cohort Size, Age, Condition	Follow-Up Period	MHT	Outcome	Ref.
PEPI RDBPC	875, 45–64 yr healthy	3 yr (1991–1994)	CEE 0.625 mg CEE+MPA 2.5 mg Oral	CEE or CEE+progestin improves lipoproteins and lowers fibrinogen levels.	[303]
HERS RDBPC	2763, 44–79 yr with CAD	4.1 yr (1993–1997)	CEE 0.625 mg +MPA 2.5 mg Oral	No overall reduction in cardiovascular events. High risk of CVD in first year.	[296]
HERS II RDBPC Follow-up	2321, 44–79 yr with CAD	6.8 (2.7 yr added to HERS) (1993–1999)	CEE+MPA Oral	MHT did not reduce risk of cardiovascular events in women with CAD.	[297]
HERS-UA (Uric Acid) RDBPC	2763, 44–79 yr	4.1 yr (1993–1997)	CEE+MPA Oral	CEE+MPA lowered serum UA levels slightly, but neither baseline UA nor change in UA affected CHD risk.	[304]
WHI RDBPC	16,608, 50–79 yr healthy 10,739 healthy with prior hysterectomy	5.2 yr (1993–1998)	CEE+MPA Oral CEE Oral	Overall health risks exceeded benefits. Overall doubling of VTE events in CEE+MPA arm; 30% to 40% increased risk of stroke.	[295]
WEST RDBPC	664, ~71 yr (46–91 yr) with prior ischemic stroke	2.8 yr (1993- 1998)	E2 Oral	No reduction in mortality or recurrence of stroke. MHT should not be used for secondary prevention of cerebrovascular disease.	[302]
PHOREA RDBPC	321, >55 yr (40–70 yr)	48 weeks (1995–1996)	E2 1 mg + gestodene 0.025 mg Oral	Did not slow progression of subclinical atherosclerosis in Post-MW at increased risk.	[305]
ERA RDBPC	309, ~65.8 yr with CAD	3.2 yr (1996–1999)	CEE CEE+MPA, Oral	Increased HDL and decreased LDL but no change in coronary atherosclerosis progression.	[306]
WISDOM RDBPC	5692, 50–69 yr	1 yr (1999- 2002)	CEE CEE+MPA, Oral	MHT increased CVD and VTE risk when started many years postmenopause.	[307]
WAVE RDBPC	423, >55 yr with coronary stenosis	2.8 yr (1999–2002)	CEE+MPA Vitamin E, C Oral	No cardiovascular benefit of MHT or vitamin C and E.	[308]
ESTHER case-control	271—case 610—control 45–70 yr	6 yr (1999–2006)	Users of E2+ Progestogen Oral versus Trans-dermal	Oral but not transdermal E2 increases VTE risk. Norpregnanes may be thrombogenic. Micronized P4 and pregnanes are safe with respect to thrombotic risk.	[309]
RUTH RDBPC	10,101, ~67.5 yr	5.6 yr (2000–2005)	Raloxifene daily Oral	Did not affect risk of CVD. Benefits in reducing invasive breast cancer and vertebral fracture should be weighed against risk of VTE and stroke.	[310]
EPAT RDBPC	222, ~57 yr healthy	2 yr (2001–2003)	E2 Oral	Slower progression of subclinical atherosclerosis.	[311]
KEEPS DBRCT	720, 42–58 yr, 6–36 months postmenopause	5 yr (2005–2010)	Oral CEE 0.45 mg/d or Transdermal E2 50 μg weekly + Oral P4 200 mg/d 12 d/month	No effect on progression of carotid intima-thickness, atherosclerosis, or accrual of coronary calcium. No CAD benefits or adverse effects.	[312,313,314]
ELITE RDBPC	643, <6 versus ≥10 yr postmenopause	2005–2012	Oral E2 1 mg/d ± Vaginal P4 gel 4%	Less progression of carotid-intima thickness and subclinical atherosclerosis when MHT was initiated within 6 yr but not ≥ 10 yr postmenopause.	[315]
DRSP+E2 RDBPC	750, 45–75 yr, with HTN	8 weeks (~2006)	DRSP 1–3 mg +E2 1 mg Oral	DRSP+E2 reduced BP in Post-MW with HTN. Decreased serum LDL-C. No increase in serum potassium.	[316]
E2+Micro P4 RDBPC	172, 45–60 yr healthy	1 yr (2010–2016)	Transdermal E2 0.1 mg patch +Micronized P4 200 mg/d oral 12 d/month	Lower diastolic BP and LDL-C. Improved cardiac autonomic baroreflex sensitivity. Prevented age-related decrease in brachial artery FMD (endothelium-dependent) and increase in stress reactivity score.	[317]
Soy ± isoflavone DBRCT	200, ~55 yr, Healthy	6 months (2012)	Oral 15 g/d soy ± 66 mg/d isoflavone for 6 months	Beneficial effect on systolic BP with soy protein plus isoflavones versus soy protein without isoflavones.	[318]

DBRCT, double-blind randomized clinical trial; DRSP, drospirenone; ELITE, Early versus Late Intervention Trial with Estradiol; EPAT, Estrogen in the Prevention of Atherosclerosis Trial; ERA, Estrogen Replacement and Atherosclerosis; ESTHER, EStrogen and THromboEmbolism Risk study; HERS, Heart and Estrogen/progestin Replacement Study; KEEPS, Kronos Early Estrogen Prevention Study; PEPI, Postmenopausal Estrogen/Progestin Interventions; PHOREA, Postmenopausal Hormone REplacement against Atherosclerosis; RDBPC, randomized double-blind placebo-controlled; RUTH, Raloxifene Use for The Heart; WAVE, Women’s Angiographic Vitamin and Estrogen trial; WEST, Women’s Estrogen for Stroke Trial; WHI, Women’s Health Initiative; WISDOM, Women’s International Study of long Duration Oestrogen after Menopause; yr, year.

**Table 3 ijms-26-05078-t003:** Changes in ER subtypes and levels in representative biological systems in Pre-MW versus Post-MW.

System	Tissue	Pre-MW	Post-MW	Function/Significance	Ref.
Eye/Vision	Retina, retinal pigment epithelium, ciliary body, iris, and lens epithelium	ERα	↓ ERα	Alterations in ER may be involved age-related macular degeneration, idiopathic full-thickness macular hole, glaucoma, cataract, and dry eye.	[348]
Cardiovascular System	Coronary artery all layers, mainly VSMCs	ERβ > ERα	No change among MHT nonusers. ↓ ERα in MHT users.	Dominant role of ERβ in coronary arteries and in association with atherosclerosis and calcification. The association of ERβ and atherosclerotic severity is not linked to age. MHT may decrease ERα expression.	[108,115]
	Aorta VSMCs	ERα ~30% ERβ ~70%	ERα ~23% ERβ 77% (MHT user)	Relative abundance of ERβ mRNA supports E2/ERβ-mediated inhibition of VSMC proliferation/migration. ERβ levels may not correlate with age.	[108]
	Peripheral arterioles from discarded adipose tissue	ERβ and GPER in ECs and ERα in adventitia	↓↓ ERβ ↓↓ GPER	Continuous E2 exposure in cis- and transgender females or Post-MW may suppress EC ERβ and GPER, causing oxidative stress, decreased FMD, and microvascular damage.	[255]
	Peripheral veins	ERα in ECs ↓ in early versus late follicular phase of menstrual cycle	↓↓ ERα	ERα levels positively related to eNOS expression, ser1177 phosphorylation, and endothelium-dependent brachial artery FMD and are modulated by fluctuations in E2 status during menopausal cycle and age-related E2 deficiency in Post-MW.	[361]
Digestive System	Buccal mucosa, minor salivary, submandibular, and parotid glands	ERs	↓ ERs	E2/ER-responsive tissues. MHT may be effective for menopausal mucosal diseases and oral discomfort.	[365,366]
	Anal canal hemorrhoid tissue	ERs	No change	No differences in anal canal ERs in relation to menopausal status or age.	[367]
Adipose Tissue	Visceral and subcutaneous	ERα>ERβ	↓ ERα ↑ ERβ	↓ ERα/ERβ ratio may affect adipose tissue metabolism in Post-MW.	[368,369,370,371]
Nervous System	Dorsolateral supraoptic nucleus	ERβ>ERα	↑ ERα ↓↓ ERβ	↓ ERβ and ↑ ERα in Post-MW may activate arginine vasopressin neurons and increase incidence of HTN and CVD.	[372]
Reproductive System	Vaginal wall and uterosacral ligaments	ERα and ERβ	↓ ERα ↓↓ ERβ	Decreased ER expression, particularly ERβ, may play a role in common symptoms of vaginal atrophy and dryness in Post-MW.	[373,374]
	Myometrium	ERβ/ERα ratio 0.6–1.5	↓ ERα ↑ ERβ ERβ:ERα ratio 2.5–7.6	Altered ERα and ERβ expression may be associated with pathological myometrium growth.	[375]
	Ovaries	ERα in thecal cells, ERβ in granulosa cells, and ERα and ERβ in interstitial gland cells	↓↓ ERβ	In Pre-MW, high ERβ expression in granulosa cells may be associated with growth and development of follicles, which are reduced in Post-MW.	[376]

↓, decreased; ↓↓, markedly decreased; ↑, increased.

**Table 4 ijms-26-05078-t004:** Guide for MHT use in Post-MW.

Menopausal Stage	Early	Perimenopause	Late
Age (years)	40–50	50–60	>60
Years since menopause	Within 10	10–15	>15
Smoking	-	+	+
Cardiovascular health			
BMI	18.5–25 (Normal)	25–30 (Overweight)	≥30 (Obese)
Physical activity	Active	Limited	Sedentary
BP (systolic/diastolic) mmHg	<120/<80 Normal	120–129/<80 Elevated	≥130/≥80 HTN
10-year ASCVD risk	<5%	5–10%	>10%
Pre-existing CVD	-	-	Congenital heart disease ASCVD/CAD/PAD Clotting disorders, VTE MI, stroke/TIA
Other pre-existing conditions			
Diabetes/metabolic syndrome	-	+	++
Lipid profile Hyperlipidemia	↑ HDL, ↓ LDL -	↓ HDL, ↑ LDL +	↓↓ HDL, ↑↑ LDL ++
Autoimmune disease	-	+	++
Breast/uterine cancer	Low risk	High risk	Detected
MHT risk	Low	Intermediate	High
MHT guide			
Route	Oral/transdermal	Transdermal preferred	Avoid oral/transdermal Topical preferred
Dose To control menopausal and genitourinary symptoms	E2 1 mg/d oral or 0.025–0.05 mg/d transdermal + Micronized P4 100–200 mg/d oral	E2 0.5 mg/d oral or 0.025 mg/d transdermal (preferred) + Micronized P4 100 mg/d oral	Vaginal E2 at minimal effective dose
Other	Transdermal E2 reduces risk of VTE/stroke	Lifestyle adjustments. Diet, exercise, lipid-lowering drugs (statins)	Consider non-hormonal treatment

ASCVD, atherosclerotic cardiovascular disease; MI, myocardial infarction; PAD, peripheral artery disease; TIA, transient ischemic attack (mini-stroke); VTE, venous thromboembolism. -, No; +, mild; ++, severe; ↑, increased; ↑↑, markedly increased; ↓, decreased; ↓↓ markedly decreased.

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
