# Peer review of "Hormone Replacement Therapy and Cardiovascular Health in Postmenopausal Women"

_ijms, 2025, doi:10.3390/ijms26115078_

Round 1
Reviewer 1 Report
Comments and Suggestions for Authors
Here are my response for the questions related to this review article:
* What is the main question addressed by the research?
This review article addresses many important areas such as role of sex hormones on cardiovascular disease risk, hormone replacement therapy in post-menopausal women, sex differences in vascular function, and sex hormones on vascular function along with some details on estrogen and testosterone induced mechanisms.
* Do you consider the topic original or relevant to the field? Does it address a specific gap in the field? Please also explain why this is/ is not the case.
Yes, this topic is very relevant and significant to the field of cardiovascular disease and influence of sex hormones. There is a large gap in understanding the role and mechanisms of sex hormones induced cardiac function and the risk involved due to depletion of sex hormones with age. Also with increased in transgender population, this topic will be of high importance.
* What does it add to the subject area compared with other published material?
This review article covers in-depth knowledge on sex hormones and cardiovascular risk.
* What specific improvements should the authors consider regarding the methodology?
This is a review article, not a research article. So, no methodology in this manuscript.
* Are the conclusions consistent with the evidence and arguments presented and do they address the main question posed? Please also explain why this is/is not the case.
Yes, conclusion and consistent with the content or scope of the review.
* Are the references appropriate?
Yes.
* Any additional comments on the tables and figures.
None.
Author Response
Responses to Reviewer 1:
* What is the main question addressed by the research?
This review article addresses many important areas such as role of sex hormones on cardiovascular disease risk, hormone replacement therapy in post-menopausal women, sex differences in vascular function, and sex hormones on vascular function along with some details on estrogen and testosterone induced mechanisms.
Response: Thank you for the encouraging comment.
* Do you consider the topic original or relevant to the field? Does it address a specific gap in the field? Please also explain why this is/ is not the case.
Yes, this topic is very relevant and significant to the field of cardiovascular disease and influence of sex hormones. There is a large gap in understanding the role and mechanisms of sex hormones induced cardiac function and the risk involved due to depletion of sex hormones with age. Also with increased in transgender population, this topic will be of high importance.
Response: Thank you for the supporting comment.
* What does it add to the subject area compared with other published material?
This review article covers in-depth knowledge on sex hormones and cardiovascular risk.
Response: Thank you for the good comment.
* What specific improvements should the authors consider regarding the methodology?
This is a review article, not a research article. So, no methodology in this manuscript.
Response: Thank you for the supporting comment.
* Are the conclusions consistent with the evidence and arguments presented and do they address the main question posed? Please also explain why this is/is not the case.
Yes, conclusion and consistent with the content or scope of the review.
Response: Thank you for the uplifting comment.
* Are the references appropriate?
Yes.
Response: Thank you.
* Any additional comments on the tables and figures.
None.
Response: Thank you very much for the good comments.
Reviewer 2 Report
Comments and Suggestions for Authors
This manuscript is very relevant to a better understanding of postmenopausal hormone therapy and cardiovascular risk.
- I was curious to know how the articles presented in the manuscript were selected. Although this is a narrative review, it is essential that the author presents the total number of articles cited, the selection method, and the databases used (PubMed? Scopus?)
- After the 2 sessions “Estrogen receptors (ERs) in the vasculature” and “E2 and ECM,” it would be very interesting if the author included an image of the action of E2 on the endothelium. This would result in a greater number of views and instructions for the article.
- In the session “Route of administration of MHT and CVD,” the author should include more information about estrogen via the vaginal route. I suggest including more data or comments on this route: DOI: 10.1093/humrep/dew014
- Still in this session “Route of administration of MHT and CVD,”, the author should create a table with the routes and doses of hormone therapies, highlighting the known benefits and harms of each formulation.
- The author mentions implants in the introduction but does not include any cardiovascular assessment studies of this approach in women. We are aware of the scarcity of studies, but some references can be cited since this therapy has been gaining prominence in the literature. An observational study evaluated some cardiovascular outcomes in 1 million procedures: DOI: 10.1177/20420188211015238 or testosterone in women doi: 10.1016/j.maturitas.2010.12.001. and DOI: 10.3390/ph16040619
- At the end of the manuscript in the “Conclusions and perspectives” section, the author tries to summarize the findings; however, the reader is confused by such a long conclusion that even includes citations. I suggest creating a “Discussion and Perspectives” section presenting all the findings and creating a new “Conclusion” section with only a short and more specific summary of the main findings.
- The author also did not include the limitations of this review; I suggest including this in the suggested “discussion and perspectives” section. This is essential since much of the content presented in the article is from studies with men and not with women. It is essential to cite this and it will bring transparency to the manuscript.
Best Wishes
Author Response
Responses to Reviewer 2:
This manuscript is very relevant to a better understanding of postmenopausal hormone therapy and cardiovascular risk.
Response: Thank you for the supporting comment.
I was curious to know how the articles presented in the manuscript were selected. Although this is a narrative review, it is essential that the author presents the total number of articles cited, the selection method, and the databases used (PubMed? Scopus?)
Response: Thank you for raising this important point. We clarified the total number of articles cited, the selection method, and the databases used at the end of the Introduction and indicated that: “As a narrative review, our goal was to synthesize key perspectives rather than exhaustively include all literature. The review cites 479 articles, selected following three steps:
- Database search: The PubMed database was searched for relevant articles published between 2000-2024 using the keywords: Hormone Replacement Therapy, Menopausal Hormone Therapy, and Cardiovascular Health. This initial research yielded ~820 articles.
- Screening: Careful screening excluded non-English articles, case reports, and small sample size research without justification, which narrowed the search to 600 articles for further review.
- Final inclusion: 479 articles were cited based on relevance, impact, expert input and novelty.
The cited articles represent the most influential and evidence-supported research work in this field.
After the 2 sessions “Estrogen receptors (ERs) in the vasculature” and “E2 and ECM,” it would be very interesting if the author included an image of the action of E2 on the endothelium. This would result in a greater number of views and instructions for the article.
Response: Thank you for your comment. In section “Estrogen Receptors (ERs) in the Vasculature”, we listed Fig. 4 which illustrates the E2/ER-mediated genomic and non-genomic pathways in endothelial cells.
In the session “Route of administration of MHT and CVD,” the author should include more information about estrogen via the vaginal route. I suggest including more data or comments on this route: DOI: 10.1093/humrep/dew014
Response: Thank you for raising this important point. We indicated in the section subtitled “Route of administration of MHT and CVD” that “A nationwide Finnish cohort study including 195,756 postmenopausal women during the period between 1994 and 2009 found that vaginal MHT could reduce the risk of death from coronary artery disease and stroke in all age groups, with the highest risk reduction in women aged 50-59 years [DOI: 10.1093/humrep/dew014].”
Still in this session “Route of administration of MHT and CVD,”, the author should create a table with the routes and doses of hormone therapies, highlighting the known benefits and harms of each formulation.
Response: Thank you for your comment. We included Table 4 “Guide for MHT use in Post-MW” highlighting the different routes of administration, dosages, and benefits vs risks of various MHT formulations.
The author mentions implants in the introduction but does not include any cardiovascular assessment studies of this approach in women. We are aware of the scarcity of studies, but some references can be cited since this therapy has been gaining prominence in the literature. An observational study evaluated some cardiovascular outcomes in 1 million procedures: DOI: 10.1177/20420188211015238 or testosterone in women doi: 10.1016/j.maturitas.2010.12.001. and DOI: 10.3390/ph16040619
Response: Thank you for raising this important point. We included further information regarding the subcutaneous implants in the section subtitled “MHT route of administration and CVD” and indicated that “Hormone replacement therapy using silastic and bioabsorbable hormone implants has gained considerable interest as a practical and effective route for treating menopausal symptoms and reduced sexual desire disorders. A retrospective study of 1,204,012 subcutaneous hormone implants during the time period between 2012 and 2019 and included 376,254 men and women patients, 85% of whom were women, 46% of women were postmenopausal, found that subcutaneous hormone implants were safer than other routes of administration of bioidentical hormone replacement therapy for both men and women [ DOI: 10.1177/20420188211015238]. Also, Pre-MW and Post-MW who reported comparable hormone deficiency symptoms showed similar improvement in total score of validated Health Related Quality of Life and Menopause Rating Scale, as well as psychological, somatic and urogenital subscale scores with T implant therapy, with better effects in women with severe complaints, and greater improvement with higher T doses (doi: 10.1016/j.maturitas.2010.12.001). Nevertheless, based on relevant and recent research, the cardiovascular risks and safety of subcutaneous hormone implants for postmenopausal women remain to be thoroughly examined [DOI: 10.3390/ph16040619].”
At the end of the manuscript in the “Conclusions and perspectives” section, the author tries to summarize the findings; however, the reader is confused by such a long conclusion that even includes citations. I suggest creating a “Discussion and Perspectives” section presenting all the findings and creating a new “Conclusion” section with only a short and more specific summary of the main findings.
Response: Thank you for your suggestion. We now included two separate sections; a more detailed “Discussion and perspectives” section, and a brief “Conclusion” section.
The author also did not include the limitations of this review; I suggest including this in the suggested “discussion and perspectives” section. This is essential since much of the content presented in the article is from studies with men and not with women. It is essential to cite this and it will bring transparency to the manuscript.
Response: Thank you for your suggestion. We discussed the limitations of the review at the end of the section subtitled “Discussion and perspectives” and indicated that “We should note some of the limitations of this review. Our inclusion criteria may have inadvertently excluded relevant studies, particularly those published in non-English language or in less accessible journals, and some pertinent information may have been missed. Also, while our intended focus was on MHT and CVD in Post-MW, a substantial amount of information was derived from studies in men or male animals. Future reviews could address these gaps by employing, broader language inclusion, and describing more large scale comprehensive studies in women and detailed experiments in female animals as they become available”.
Best Wishes
Response: Thank you and best wishes to you too.
Reviewer 3 Report
Comments and Suggestions for Authors
This manuscript offers a thorough and insightful review of the complex relationship between menopausal hormone therapy and cardiovascular health. Authors provide a comprehensive and mechanistically detailed review of the effects of sex hormones on vascular function, with a particular focus on menopausal hormone therapy and its implications for cardiovascular health in postmenopausal women. However, while the breadth and scientific rigor are commendable, the article would benefit from improvements in structural clarity, synthesis of clinical relevance, and methodological transparency in literature selection.
The manuscript is highly informative, but major revisions are recommended to improve structural clarity, integrate more practical clinical guidance, and specify the methodology for literature selection.
Author Response
Responses to Reviewer 3:
This manuscript offers a thorough and insightful review of the complex relationship between menopausal hormone therapy and cardiovascular health. Authors provide a comprehensive and mechanistically detailed review of the effects of sex hormones on vascular function, with a particular focus on menopausal hormone therapy and its implications for cardiovascular health in postmenopausal women. However, while the breadth and scientific rigor are commendable, the article would benefit from
Response: Thank you very much for the constructive comment. We carefully revised the manuscript and provided further improvements in structural clarity, synthesis of clinical relevance, and methodological transparency in literature selection.
The manuscript is highly informative, but major revisions are recommended to improve structural clarity, integrate more practical clinical guidance, and specify the methodology for literature selection.
Response: Thank you for your suggestion. We specifically clarified the total number of articles cited, the selection method, and the databases used at the end of the Introduction and indicated that: “As a narrative review, our goal was to synthesize key perspectives rather than exhaustively include all literature. The review cites 479 articles, selected following three steps:
- Database search: The PubMed database was searched for relevant articles published between 2000-2024 using the keywords: Hormone Replacement Therapy, Menopausal Hormone Therapy, and Cardiovascular Health. This initial research yielded ~820 articles.
- Screening: Careful screening excluded non-English articles, case reports, and small sample size research without justification, which narrowed the search to 600 articles for further review.
- Final inclusion: 479 articles were cited based on relevance, impact, expert input and novelty.
The cited articles represent the most influential and evidence-supported research work in this field.”
Round 2
Reviewer 2 Report
Comments and Suggestions for Authors
The authors answered all my questions appropriately. Best Wishes